# Functionally distinct roles for eEF2K in the control of ribosome availability and p-body abundance

Patrick R. Smith [1,5], Sarah Loerch [2,3,5], Nikesh Kunder[1], Alexander D. Stanowick [1], Tzu-Fang Lou[1] &
Zachary T. Campbell [1,4✉]

Processing bodies (p-bodies) are a prototypical phase-separated RNA-containing granule. Their abundance is highly dynamic and has been linked to translation. Yet, the molecular mechanisms responsible for coordinate control of the two processes are unclear. Here, we uncover key roles for eEF2 kinase (eEF2K) in the control of ribosome availability and p-body abundance. eEF2K acts on a sole known substrate, eEF2, to inhibit translation. We find that the eEF2K agonist nelfinavir abolishes p-bodies in sensory neurons and impairs translation. To probe the latter, we used cryo-electron microscopy. Nelfinavir stabilizes vacant 80S ribosomes. They contain SERBP1 in place of mRNA and eEF2 in the acceptor site. Phosphorylated eEF2 associates with inactive ribosomes that resist splitting in vitro. Collectively, the data suggest that eEF2K defines a population of inactive ribosomes resistant to recycling and protected from degradation. Thus, eEF2K activity is central to both p-body abundance and ribosome availability in sensory neurons.

[1] The University of Texas at Dallas, Department of Biological Sciences, Richardson, TX, USA. [2] Janelia Research Campus, Howard Hughes Medical Institute, Ashburn, VA, USA. [3] University of California, Santa Cruz, Department of Chemistry and Biochemistry, Santa Cruz, CA, USA. [4] The Center for Advanced Pain Studies (CAPS), University of Texas at Dallas, Richardson, TX, USA. [5]These authors contributed equally: Patrick R. Smith, Sarah Loerch. ✉email: Zachary.Campbell@utdallas.edu

RNA control permeates biology. Every aspect in the brief life of an mRNA is meticulously controlled by proteins. Protein-RNA complexes can assemble into large biomolecular condensates[1]. Some form microscopically visible granules or membraneless organelles that have been implicated in transcription, mRNA stability, localization, and translation[2,3]. Their assembly can be highly dynamic and responsive to an array of cell autonomous and non-autonomous signaling events[4–6]. Translation and the activity of ribosomes are intimately linked to the abundance of multiple RNP granules[7]. Understanding the regulatory events that bridge granule dynamics to translation is of fundamental importance.

P-bodies are an archetypal membraneless organelle. They are enriched for proteins linked to mRNA metabolism and poorly translated mRNAs[3,8,9]. P-bodies are not major sites of RNA metabolism as their loss has negligible effects on RNA decay[10]. Furthermore, decay intermediates are absent from p-bodies[11–13]. It is hypothesized that they function as storage sites of translationally repressed mRNAs[8]. While their biological functions remain unclear, critical insights have emerged into the factors that govern their formation.

A broad set of cues impact p-body assembly. In *S. cerevisiae*, glucose deprivation, activation of protein kinase A, and osmotic stress promote formation of p-bodies[14–16]. In mammals, their abundance can differ substantially between cell types. Neurons are exemplary. They possess approximately an order of magnitude more than immortalized cell lines under basal conditions[17]. Intriguingly, signaling molecules that promote persistent changes in neuronal plasticity can also modulate p-body number and distribution[17–19]. For example, stimulation of metabotropic or ionotropic glutamate receptors (mGluRs or NMDARs), results in reduced p-body abundance in dendrites of hippocampal neurons[20]. It is unclear if the molecular mechanisms responsible for the control of p-bodies are similar between immortalized cell lines and sensory neurons.

Translation has been linked to p-body dynamics[7]. This relationship has been studied extensively in mitotically active mammalian cell lines. Perturbation of translation initiation increases p-bodies and cytoplasmic mRNA[21]. Similarly, premature translation termination with puromycin, which indirectly promotes release of mRNA, increases the number of p-bodies[10,22]. Trapping mRNAs on polysomes with the elongation inhibitor cycloheximide reduces p-bodies[23,24]. A corollary of these observations is that mRNA might be limiting for p-body assembly[7]. A major focus of this work is investigating the generality of this model.

During translation, ribosomes decode mRNAs to produce proteins. Prior to translation initiation, the 40S and 60S ribosomal subunits assemble on mRNA to form an 80S ribosome. During peptide chain elongation, intersubunit rotations facilitate the translocation of the tRNA–mRNA module, which is coupled to the nascent polypeptide. After the elongation phase is completed, ribosomes are recycled by splitting of the 80S into individual subunits[25,26]. However, 80S ribosomes can exist stably in the absence of mRNA. In *S. cerevisiae*, starvation induces formation of 80S ribosomes that contain the hibernation factor Stm1p (SERBP1 in mammals) in the mRNA channel and eEF2 in the A site[27]. Stm1p aids in cellular recovery after starvation stress and promotes resumption of translation[28–30]. While compositionally similar ribosomes are broadly conserved in metazoans, the signaling events that mediate their assembly and recycling, and their role in the translation cycle remain opaque[27,31–33].

A prominent mechanism of translational control is regulation of elongation by the Eukaryotic elongation factor 2 kinase (eEF2K). Eukaryotic elongation factor 2 (eEF2) promotes translocation of elongating ribosomes[34,35]. eEF2K catalyzes phosphorylation of eEF2 at Thr56, which inhibits translation[34–37].

Although the precise mechanism is unclear, phosphorylation might incapacitate binding to actively translating ribosomes[37]. eEF2K is controlled by a broad range of upstream signaling pathways, and has been linked to a range of key neuronal processes including synaptic plasticity, learning, and memory[38–46]. For example, NMDA-type ionotropic glutamate receptors (NMDARs) have established roles in plasticity and stimulate eEF2K activity[47–52].

Here, we sought to examine the relationship between translation and p-body dynamics in mouse sensory neurons isolated from the dorsal root ganglion (DRG). We found that, in contrast to mitotic cells, multiple inhibitors of protein synthesis failed to affect the abundance of sensory neuron p-bodies (SNPBs). However, enhancement of eEF2K activity with the HIV protease inhibitor nelfinavir resulted in a near loss of SNPBs and a reduction in translation. Nelfinavir caused a reduction of polysomes and a substantial accumulation of 80S ribosomes. Single molecule cryo-electron microscopy revealed ribosomes bound to eEF2 in the acceptor site, and SERBP1 in the mRNA channel. Subsequent structural and biochemical investigation revealed phosphorylated eEF2 on purified 80S ribosomes. Finally, vacant ribosomes formed after addition of nelfinavir are resistant to splitting. Our experiments reveal that eEF2K plays distinct roles in the regulation of SNPB dynamics and ribosome availability.

## Results

**Effects of translation inhibitors on SNPB abundance.** We first examined the relationship between translation and p-bodies in sensory neurons, using an array of small molecules. Homoharringtonine blocks the first translocation step after recruitment of the large subunit to the pre-initiation complex[53,54]. Puromycin causes dissociation of the nascent peptide chain and ribosomal subunits[55,56]. Cycloheximide disrupts translocation of A- and P-site tRNAs by binding to the E site of the large subunit[57–59]. Emetine blocks elongation by binding to the E site of the small subunit[60,61]. Notably, emetine inhibits translocation of the mRNA–tRNA module but does not inhibit intersubunit rotation. Ribosomes treated with emetine are trapped in a hybrid state where the peptidyl-tRNA is in the A/P configuration and likely can accommodate eEF2[56,62].

To determine the effects of protein synthesis inhibitors on SNPB abundance, we conducted immunocytochemistry (ICC). As a marker of the SNPBs, we used RCK/Ddx6 (Fig. 1A)[19,63–65]. Primary DRG cultures contain non-neuronal cells that facilitate neuronal viability. To measure SNPBs specifically in neurons, we co-labeled with a neuronal marker (peripherin). Neurons averaged 64 SNPBs per cell. Homoharringtonine (Sigma), puromycin (ThermoFisher), and cycloheximide (Sigma) did not affect SNPB abundance. However, emetine (Sigma) led to a modest reduction in SNPBs. As a comparison, we repeated these treatments in U2-OS cells. They are commonly used to study cytoplasmic membraneless organelles[13,21,66]. In agreement with prior findings in mitotic cell lines, puromycin resulted in an increase in p-body number, while arrest of polysomes with cycloheximide or emetine resulted in a loss of p-bodies (Fig. 1B)[10,22–24]. Interestingly, runoff of translating ribosomes with homoharringtonine also lead to a loss of p-bodies.

Though the inhibitors used have well established effects on translation, we nonetheless sought to exclude the unlikely possibility that these exhibit altered effects on translation in neurons. We measured nascent protein synthesis using metabolic pulse chase of a non-canonical amino acid, an approach termed fluorescent non-canonical amino acid tagging (FUNCAT). In this assay, cells are allowed to incorporate a methionine analogue, L-azidohomoalanine (AHA), which is later covalently labeled

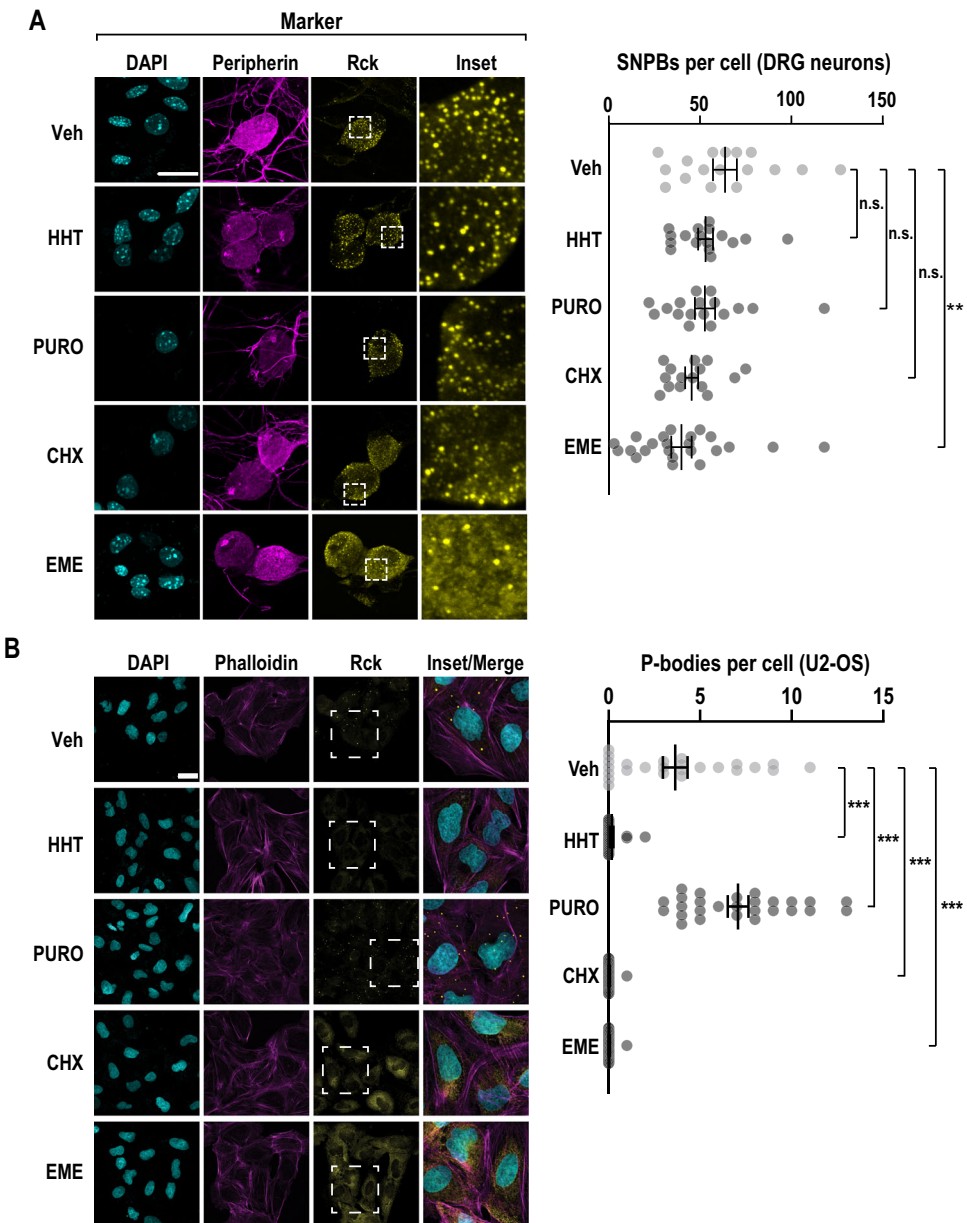

**Fig. 1 The translation inhibitor emetine reduces p-bodies in primary sensory neurons. A** Primary DRG cultures were treated with vehicle (Veh), homoharringtonine (HHT, 50 µM), puromycin (PURO, 10 µM), cycloheximide (CHX, 20 µg/ml), or emetine (EME, 50 µM) for a period of 1 h and subjected to ICC. Confocal imaging was used to identify p-bodies and key markers. DRG neurons were identified by peripherin immunofluorescence (magenta) and SNPBs were identified based on Rck (yellow). Nuclei were stained with DAPI (cyan). **A** left Representative confocal images. Scale bar = 20 µm. **A** right Quantification of p-bodies in primary DRG neurons. The error bars represent mean ± S.E.M. For Veh, HHT, PURO, CHX, and EME $n = 17, 17, 17, 15$, and 23 cells, respectively $p$-values determined by one-way ANOVA. Veh vs EME $p = 0.0076$. **B** U2-OS cells were subjected to the same treatments as in **A** and subjected to ICC. Cells were labeled with phalloidin-TRITC (magenta) and Rck used as a marker for p-bodies (yellow). Nuclei were stained with DAPI (cyan). **B** left Representative confocal images. Scale bar = 30 µm. **B** right Quantification of p-bodies per cell. The error bars correspond to the mean ± S.E.M. For Veh, HHT, PURO, CHX, and EME, $n = 25, 24, 27, 28$, and 29 cells, respectively $p$-values determined by one-way ANOVA. Veh vs. HHT $p < 0.0001$, Veh vs. PURO $p < 0.0001$, Veh vs. CHX $p < 0.0001$, Veh vs. EME $p < 0.0001$. Source data are provided as a Source Data file.

with a fluorescent dye[67,68]. The relative amount of fluorescence was used as a proxy for the level of nascent translation, normalized to AHA-free cells. As expected, each translation inhibitor resulted in a substantial reduction in nascent protein synthesis (Fig. S1A). Thus, the failure of SNPBs to respond to translation inhibitors cannot be attributed to cell type-specific effects on translation. Taken together, these results suggest that the coupling of translation and p-bodies is fundamentally different in sensory neurons as compared to mitotic cell lines, and that the connection between translation and SNPBs is more

nuanced than expected. Based on the finding that emetine results in a significant decrease in SNPBs, we reasoned that factors involved in elongation might play critical roles in coordinate regulation of translation and SNPBs.

**Pharmacological activation of eEF2K causes loss of SNPBs**. To investigate how SNPB abundance is controlled, we focused on the elongation phase of translation. Due to emetine's unique effect on ribosome conformation, we asked if eEF2 plays a role in SNPBs.

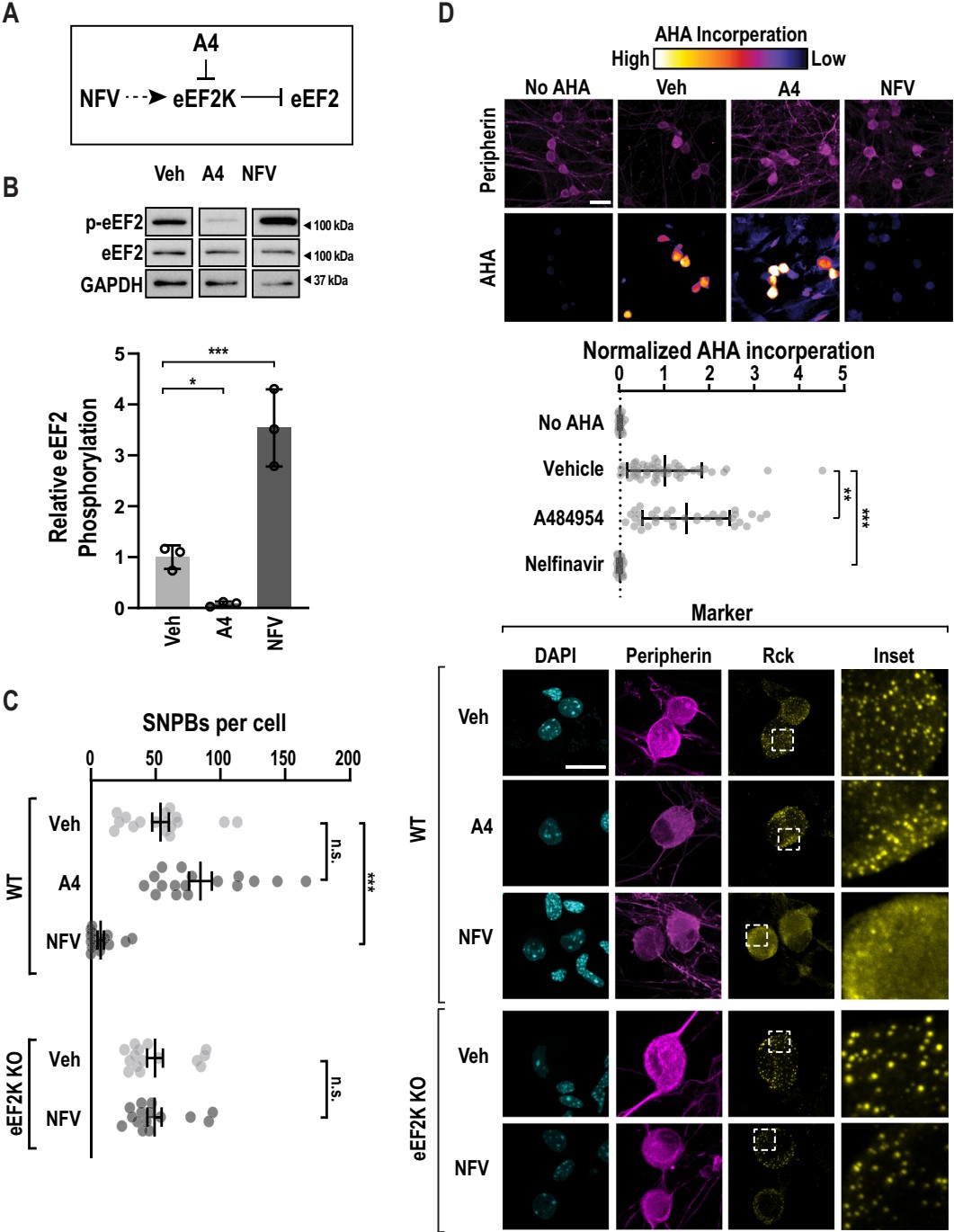

**Fig. 2 eEF2K controls p-body numbers in sensory neurons. A** A schematic depicting the effects of an eEF2K inhibitor, A484594 (A4), or an eEF2K activator, Nelfinavir (NFV) on eEF2K and eEF2. **B** Primary DRG cultures were again treated with vehicle (Veh), A484954 (A4, 25 μM), or nelfinavir (NFV, 50 μM). Lysates from treated cells were probed for p-eEF2, eEF2, and GAPDH (load control). **D** upper Representative immunoblots (cropped to depict one representative band per condition). **D** lower Quantification represents mean p-eEF2/eEF2 signal, $n = 3$ biological replicates. Error bars represent ±SD. $P$-values determined by one-way ANOVA. Veh vs. A4 $p = 0.0493$, Veh vs. NFV $p = 0.0010$. **C** Primary DRG cultures were treated with vehicle (Veh), an eEF2K inhibitor, A484594 (A4), or an eEF2K agonist, Nelfinavir (NFV) for a period of 1 h. As a specificity control, NFV, was applied to DRG neurons obtained from eEF2K homozygous loss of function animals ($n = 15$). As before, peripherin (magenta) and Rck (yellow) immunofluorescence were used to identify neurons and SNPBs, respectively. **A** left Representative confocal images. Scale bar = 20 μm. **A** right Quantification of SNPBs in peripherin-positive cells. For Veh, A4, and NFV $n = 17, 17,$ and 16 cells, respectively. The error bars represent mean ± S.E.M. $p$-values determined by one-way ANOVA. Veh vs. NFV $p < 0.0001$. **D** Primary WT DRG cultures were treated as in **B** with the addition of a 30-minute pulse of AHA. As before, cells were subjected to FUNCAT and peripherin immuno-labeling and imaged via confocal microscopy. To quantify the baseline, a control group without AHA was imaged. **C** upper Representative confocal images. Scale bar = 30 μm. **D** lower Quantification of mean AHA incorporation in peripherin-positive cells, normalized to signal from AHA-free cells. For No AHA, Veh, A4, and NFV $n = 30, 53, 36,$ and 28 cells, respectively. Bars indicate mean ± SD $p$-values determined by one-way ANOVA. Veh vs. A4 $p = 0.0016$, Veh vs. NFV $p < 0.0001$. Source data are provided as a Source Data file. Uncropped blots are presented in Supplemental Fig. 7.

Nelfinavir is an FDA-approved drug that inhibits the HIV protease. At high concentration, it is also a potent eEF2K agonist (Fig. 2A, B), though its mechanism of eEF2K activation is unclear[69,70]. We also made use of a highly specific inhibitor of eEF2K, A484954[71]. Treatment of primary DRG neurons with A484954 (Sigma) did not lead to a significant change in SNPBs (Fig. 2C). However, the eEF2K agonist, nelfinavir (Cayman Chemical), induced a near loss of p-bodies. We next interrogated the specificity of this effect with eEF2K knockout mice[72,73]. DRG neurons isolated from homozygous eEF2K KO animals show similar abundance of SNPBs to WT neurons. However, nelfinavir had no effect on SNPBs in eEF2K KO neurons (Fig. 2C). We conclude that eEF2K is not required for the formation of SNPBs, yet it plays a critical role in their regulation.

We next asked if increased eEF2K activity attenuates translation. We again quantified nascent translation using FUNCAT. eEF2K inhibition led to a slight but significant increase in translation. Conversely, nelfinavir induced a 20-fold decrease in translation in WT neurons (Fig. 2D). We determined if translational repression by nelfinavir is eEF2K-dependent using sensory neurons obtained from eEF2K deficient mice. We found that AHA incorporation was reduced by only 7-fold in eEF2K KO neurons (Fig. S1B). This suggests that that nelfinavir represses translation in part through eEF2K.

**eEF2K modulation has no impact on p-bodies in cell lines.** Neuronal p-bodies are compositionally distinct from their somatic counterparts and undergo dynamic changes in response to neurotropic growth factors and signaling molecules[17,19,20,74]. We asked if eEF2K is involved in p-body dissolution in non-neuronal cells. Surprisingly, nelfinavir resulted in an increase in PB abundance, while A484954 had no effect in U2-OS cells (Fig. S2A, B). To probe the effects of nelfinavir on eEF2 and translation, we assessed both translation and eEF2 phosphorylation. We performed FUNCAT on U2-OS cells and found that, as with primary neurons, nelfinavir significantly reduces translation (Fig. S2C). Furthermore, immunoblots confirmed the predicted effects of nelfinavir and A484954 on eEF2 phosphorylation (Fig. S2D, E). We conclude that the effects of compounds that modulate eEF2K activity on p-bodies in an immortalized cell line differ from compositionally similar condensates in primary murine sensory neurons.

**eEF2K does not co-localize with SNPBs.** Next, we sought to determine if eEF2K is expressed in DRG neurons. We analyzed previously reported single cell sequencing data (Fig. 3A)[75,76]. We first grouped cells into clusters based on principle component analysis and expression of the following marker transcripts: *Vim* (non-neuronal), *Calca* (peptidergic), *Mrgprd* (non-peptidergic), *Th* (tyrosine hydroxylase), and *Nefh* (large diameter neurons)[76]. eEF2K is detected in all cell types present in the dataset (Fig. 3B). It is most often expressed in large diameter neurons (Fig. 3C). Expression was observed more often in neurons than in non-neurons (Fisher's exact test, $p = 0.02$). To determine if eEF2K is translated in DRG neurons, we performed immunocytochemistry (Fig. 3D). We found that eEF2K forms distinct puncta in both soma and axons but is absent from the nucleus (Fig. 3E). We observed negligible co-localization between eEF2K and SNPBs (Fig. 3F). In contrast to the single cell data, we found eEF2K was present in all of the neurons we examined. A potential cause of this discrepancy is that the limited read depth in single-cell experiments underestimates the abundance of lowly expressed transcripts[77]. Collectively, these results indicate eEF2K is present in DRG neurons but does not interact directly with SNPBs.

**Rescue of SNPB loss by an NMDAR antagonist.** The activity of eEF2K is controlled by multiple pathways. We focused on NMDA-type ionotropic glutamate receptors (NMDARs) given their high level of expression in DRG neurons and established roles in plasticity[47,48]. NMDARs have been linked to p-bodies in cortical neurons, although the underlying mechanism is unclear[17,18,20]. NMDAR activation is also known to facilitate stimulation of eEF2K activity[49–52]. To determine if NMDARs regulate SNPB abundance, DRG neurons were treated with vehicle or MK801 (Selleckchem) (Fig. 4A). MK801 is a non-competitive NMDAR antagonist and reduces eEF2K activity[78]. MK801 had little effect on SNPBs. However, co-treatment of MK801 and nelfinavir restored SNPBs to normal levels (Fig. 4B). This result suggests that NMDAR inactivation rescues the repressive effects of nelfinavir on SNPBs. To determine the molecular basis for the epistatic effect of MK801 on nelfinavir, we examined eEF2 abundance and phosphorylation with immunoblots. Co-treatment of nelfinavir and MK801 reduced eEF2 phosphorylation relative to nelfinavir alone (1.3-fold increase versus ~3.5 fold with nelfinavir alone Figs. 2C and 4C). Curiously, addition of both compounds led to an increase in total eEF2 levels by an unknown mechanism (Fig. 4C). Next, we asked if NMDAR inhibition modulates translation. MK801 promotes phosphorylation of the initiation factors eIF4E and eIF4EBP through the MAPK and mTOR pathways, respectively[79,80]. Accordingly, MK801 resulted in a 2-fold increase in translation (Fig. 4D). Co-treatment with MK801 and nelfinavir led to a modest increase in translation relative to nelfinavir alone. Collectively, our observations suggest that the activity of glutamate receptors can modulate SNPB dynamics in sensory neurons.

**eEF2K activity leads to accumulation of idle ribosomes.** To determine how eEF2K activity regulates translation, we examined the effects of nelfinavir on ribosomes. Phosphorylation of eEF2 by eEF2K attenuates elongation, reportedly by preventing its interaction with the ribosome[34,36,37]. Pharmacological inhibition of elongation with cycloheximide or emetine results in stabilized polysomes[81]. A priori, arrest of elongation through eEF2K-mediated association of phosphorylated eEF2 could stall translating ribosomes resulting in an increase in polysomes. To test this idea, we performed polysome profiling using a neuronal cell line derived from DRG (F11). This was necessary to obtain sufficient material for biochemical assays. Contrary to our expectations based on small molecule elongation inhibitors, we found that nelfinavir diminished the polysome population, while the 80S population was substantially increased (Fig. 5A, orange line). This accumulation was unaffected by the removal of cycloheximide from the assay (Fig. S3A). The nelfinavir-induced accumulation of monosomes was reduced in cells pre-treated with A484954 (Fig. 5A, blue line). This suggests that eEF2K is largely responsible for the formation of monosomes induced by nelfinavir. To probe the mechanism underlying monosome accumulation, we asked if phosphorylated eEF2 interacts with ribosomes. Ribosomes were purified following treatment with nelfinavir using sucrose cushions. We found that nelfinavir treatment resulted in accumulation of phosphorylated eEF2 in pellets containing ribosomes (Fig. 5B). As loading controls, we made use of RPL5 and RPS6 as markers of the large and small subunits, respectively. To assess the cleanliness of the preparations, we conducted two key controls. In the first, we examined the pellets for the presence of a transcription factor, ATF4. It did not co-purify with ribosomes. Additionally, we disrupted the 80S ribosome through the addition of the metal chelator EDTA. We found that addition of EDTA led to the loss of the ribosome-interacting factors SERBP1 and eEF2 in ribosome pellets. We next examined eEF2

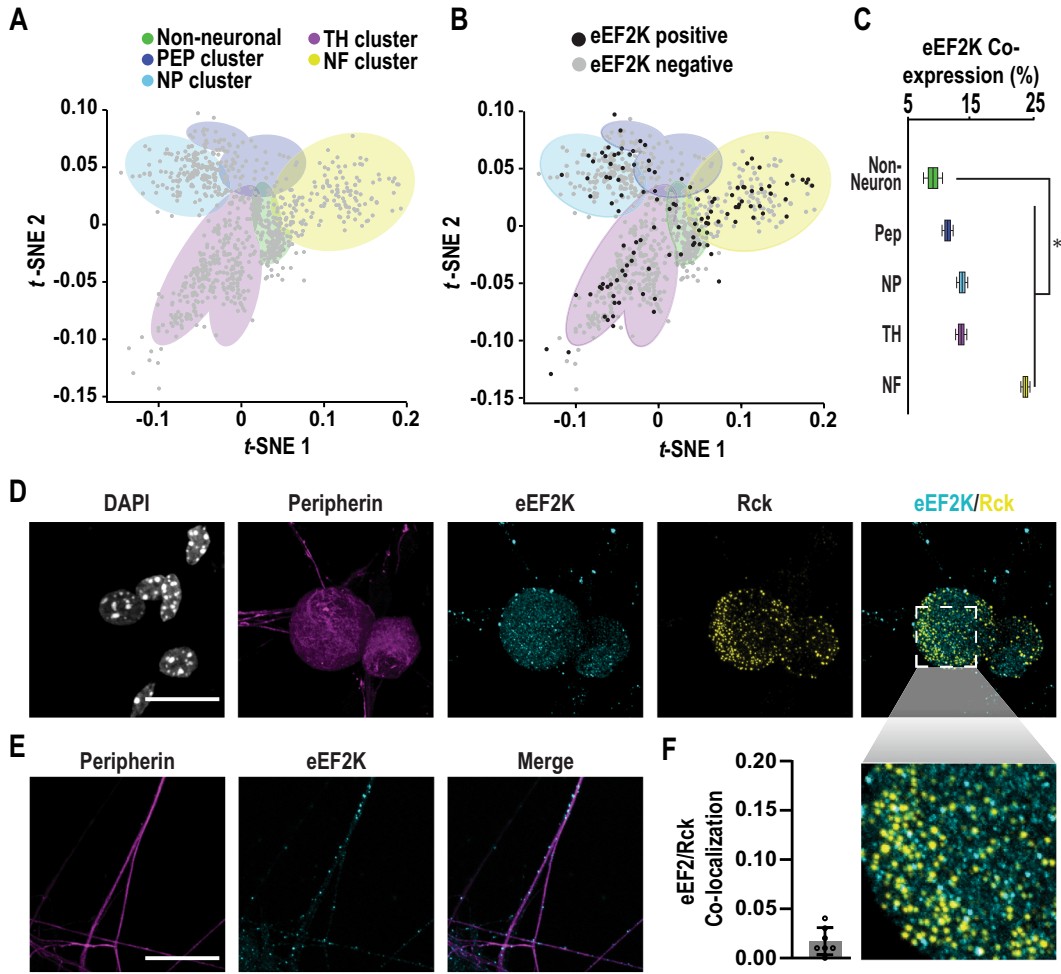

**Fig. 3 eEF2K is expressed in sensory neurons but does not localize to SNPBs. A** Single cell clusters based on expression of marker genes for the following populations of cells: non-neuronal (*Vim*), peptidergic (*Calca*/CGRP), non-peptidergic (*Mrgprd*), tyrosine hydroxylase (*Th*), and a light chain neurofilament expressed in large diameter neurons (*Nefh*). Data were obtained from Usoskin and colleagues and subjected to unbiased clustering[75,76]. **B** eEF2K expression in individual cells within these clusters. **C** Quantification of co-expression of eEF2K with marker transcripts. Boxes display mean, first, and third quartiles. Whiskers indicate minimum and maximum values. *P* value determined by Fisher's exact test, one tailed, significance of 0.05. p = 0.0357. For non-neuronal, PEP, NP, TH, and NF clusters n = 88, 73, 141, 280, and 167 cells, respectively. For details on analysis, see *Methods*. **D** Untreated primary DRG cultures were used for ICC and imaged with confocal microscopy. Representative images of cells labeled for peripherin (magenta), eEF2K (cyan), and Rck (yellow). Nuclei were stained with DAPI (white). Scale bar = 20 μm. **E** Axons of cells from **D** were also imaged. Representative confocal images show peripherin (magenta) and eEF2K (cyan) immunofluorescence signals within axons. Scale bar 20 μm. **F** Mean colocalization of eEF2K and Rck immunofluorescence signals within peripherin-positive cell bodies using Pearson's correlation coefficient (PCC), n = 7 cells. Mean Pearson's = 0.017. Error bars indicate ±SD. Source data are provided as a Source Data file.

phosphorylation status on ribosomes purified from primary DRG cultures. We found that nelfinavir induced co-purification of phosphorylated eEF2 with ribosomes similar to results obtained in F11 cells (Fig. S3B). We conclude that phosphorylation of eEF2 does not incapacitate its binding to ribosomes.

We next asked how phosphorylated eEF2 interacts with the ribosome. We treated primary DRG neurons with nelfinavir and examined purified ribosomes using cryo-EM. To exclude the possibility that ribosomal complexes become inactivated during purification, a potential consequence of high-speed centrifugation[82,83], we adopted a rapid purification method. Similar to sucrose cushion ultracentrifugation, phosphorylated eEF2 was retained on ribosomes following nelfinavir treatment (Fig. S3C). We collected a 193,693-particle dataset and used multiple rounds of maximum-likelihood classification to resolve eEF2-containing species (for classification scheme see Fig. S4A, for statistics see Table S1). The resulting reconstructions included two distinct classes with eEF2·GDP density in the ribosomal A

site, SERBP1 threaded through the mRNA channel, and E-site tRNA (Fig. 5C). Classes I and II reached resolutions of 3.1 and 3.3 Å, respectively (Fig. S4B). Overall, eEF2-bound ribosomes make up ~71% of all intact 80S ribosomes in the sample. Other classes of intact 80S ribosomes include ribosomes with E-site tRNA only. Yet, none of the classes have clear mRNA density or P-site tRNA, suggesting that actively translating ribosomes are largely absent after treatment with nelfinavir. Compositionally similar eEF2-containing complexes have been observed across different eukaryotic species including *H. sapiens* (human)[31], *S. scrofa* (pig)[32], *O. cuniculus* (rabbit)[33], *D. melanogaster* (fruit fly)[31], and *S. cerevisiae*[31,27]. In both structures (classes I and II), SERBP1 is threaded through the mRNA channel and contacts the eEF2 diphthamide (DPP715) modification site in domain IV (Fig. 5C). Consequently, this species is not a paused polysome but rather represents an 80S species that requires recycling before ribosomal subunits can participate in translation again. While most of the previous SERBP1/Stm1-containing structures are in

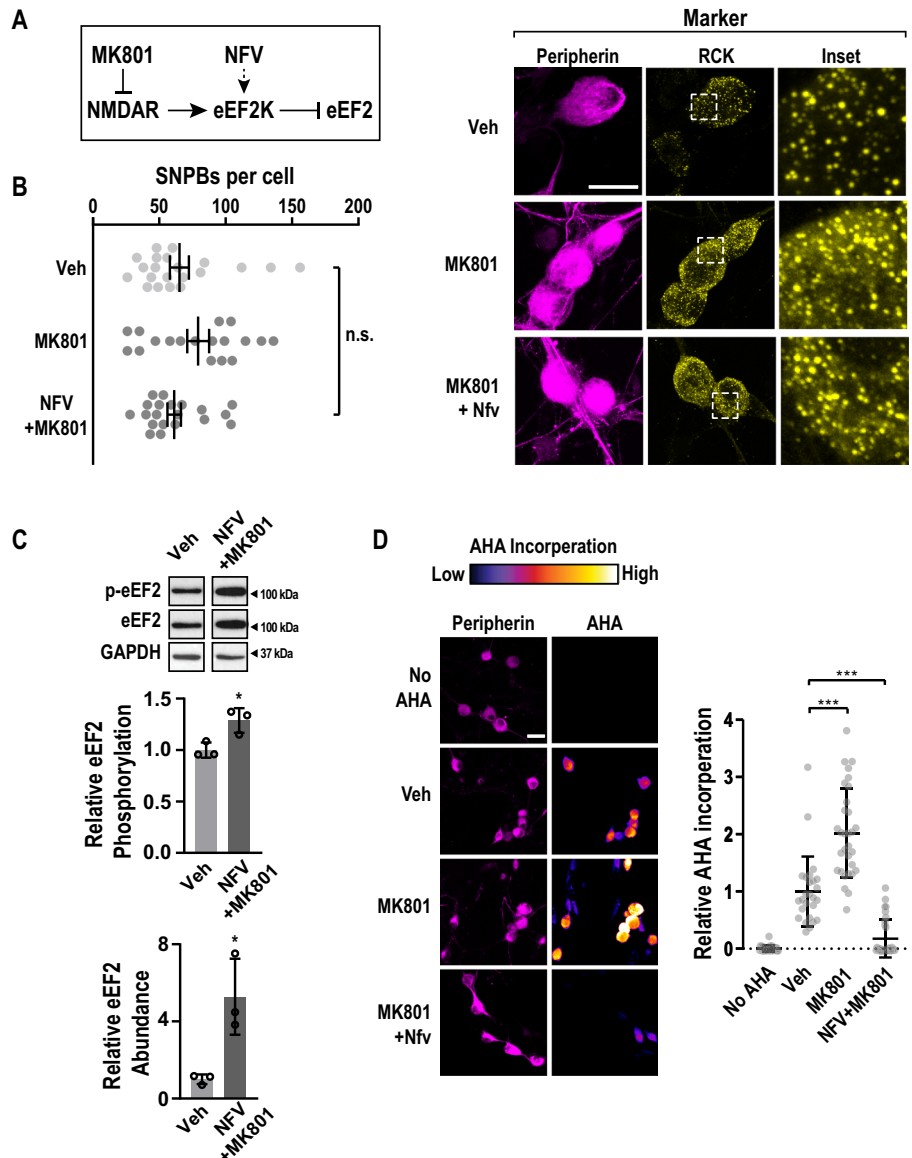

**Fig. 4 Inhibition of NMDA receptors counteracts nelfinavir-induced PB loss. A** A schematic that indicates the relationship between MK801, an NMDAR antagonist, Nelfinavir (NFV), eEF2K, and eEF2. **B** left Primary DRG cultures were treated with vehicle (Veh), MK801 (10 μM), or co-treated with nelfinavir and MK801 (NFV+MK801) for a period of 1 h, and subjected to ICC. Confocal imaging was used to identify p-bodies and key markers. DRG neurons were identified by peripherin immunofluorescence (magenta) and SNPBs were identified based on Rck (yellow). Nuclei were stained with DAPI (cyan). **B** left Representative confocal images. Scale bar = 20 μm. **B** right Quantification of PBs per peripherin-positive cell. For Veh, MK801, and MK801 + NFV, n = 21, 18, and 20 cells, respectively. Bars indicate mean ± SEM. P-values determined by one-way ANOVA. **C** Primary DRG cultures were treated with vehicle (Veh) or co-treated with nelfinavir and MK801 (NFV+MK801). Lysates from treated cells were probed for p-eEF2, eEF2, and GAPDH (load control). **C** upper Representative immunoblots (cropped to depict one representative band per condition). **C** middle Quantification of mean p-eEF2/eEF2 signal. **C** lower Quantification of mean eEF2 normalized to GAPDH, n = 3 biological replicates. Error bars represent ±SD. P-values determined by Unpaired two-tailed t test. Upper panel p = 0.329, lower panel p = 0.0204. **D** Primary DRG cultures were subjected wo the same treatments as in **B** with the addition of a 30-min pulse of AHA. Cultures were then used for FUNCAT as well as peripherin immune-labeling before confocal imaging. As a control, a no-AHA group was also imaged. **D** left Representative confocal images. Scale bar = 30 μm. **D** right Quantification of AHA incorporation in peripherin positive cells. For No AHA, Veh, MK801, and MK801 + NFV, n = 29, 26, 31, and 31 cells, respectively. Bars indicate mean ±SD. P-values determined by one-way ANOVA. Veh vs. MK801 p < 0.0001, Veh vs. MK801 + NFV p < 0.0001. Source data are provided as a Source Data file. Uncropped blots are presented in Supplemental Fig. 7.

the rotated state, similar to class I, we also identify a non-rotated conformation, which are virtually identical to those observed in rabbit reticulocyte lysate[33].

Neither class I nor II fully agrees with any of the known functional states of canonical translation. During translation, the ribosome undergoes a sequence of intersubunit rearrangements. PRE- and POST-states describe conformations observed before and after translocation, respectively, and conversion proceeds via several translocation intermediate (TI)-states. Each state is characterized by specific 40S head and body conformations[84,85]. Both classes have approximately the same extensive head swivel (15° head swivel compared to the classical PRE-1 state, PDB ID: 6y0g) but they differ in 40S body rotation. Class I is in the rotated state (4° rotation compared to the classical

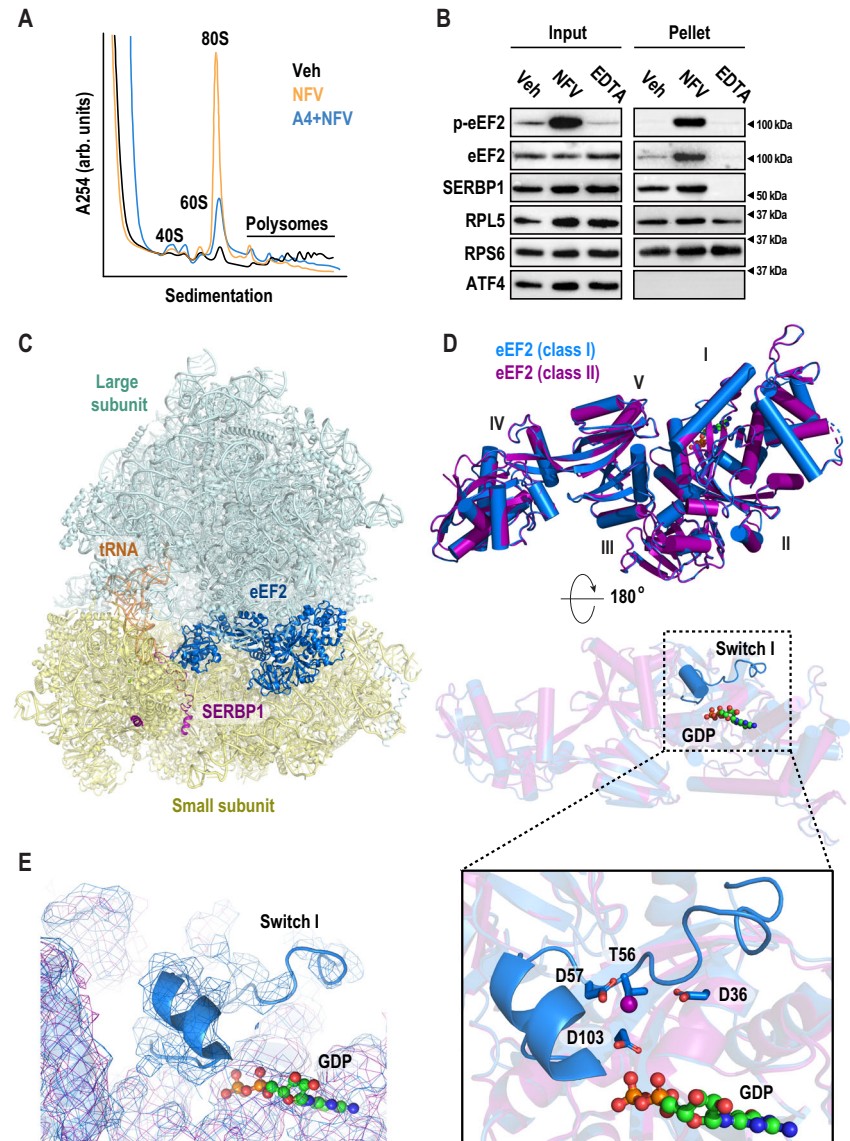

**Fig. 5 Nelfinavir treatment inactivates ribosomes via eEF2. A** Representative polysome profiles following treatment with vehicle (black) or nelfinavir (red). F11 cells were treated with either vehicle (Veh) or nelfinavir (NFV) for 1 h, followed by 5 min of cycloheximide (CHX, 100 μg/ml). Cells were lysed and used to generate polysome profiles. **B** Representative immunoblots from three biological replicates of ribosomes purified by sucrose cushion (cropped to depict one representative band per condition). F11 cells were treated with either vehicle (Veh) or nelfinavir (NFV) for 1 h, followed 100 μM emetine for 5 min. Cells were lysed and loaded on 30% sucrose cushions before ultracentrifugation to pellet ribosomes. An additional vehicle sample was further treated with EDTA (30 μM) to dissociate polysomes prior to loading on sucrose cushion. Immunoblots were performed using input and resuspended ribosome pellets. **C** Cryo-EM structure of eEF2-bound 80S mouse ribosome in the rotated state. The large subunit (LSU) is shown in pale cyan, the small subunit in pale yellow (SSU), eEF2 in marine, SERBP1 in purple, and E-site tRNA in orange. **D** Overlay of eEF2 (marine) and p-eEF2 (purple) structures shows nearly identical conformations (RMSD 0.256 Å² for corresponding $C_{alpha}$ atoms). The two structures differ in the presence of an ordered switch I in the unphosphorylated eEF2 (marine) near the bound GDP. **E** Density of eEF2 (marine) and p-eEF2 shows the presence of switch I in the unphosphorylated eEF2 structure. **F** eEF2 Thr56, target of eEF2K, is surrounded by negatively charged residues Asp57, Asp36, and Asp103 and is oriented towards the GDP beta-phosphate. This suggests that switch I rearranges upon Thr56-phosphorylation due to electrostatic repulsion. Uncropped blots are presented in Supplemental Fig. 7.

PRE-1 state, PDB ID: 6y0g) and class two is in a back-rotated conformation (4° back-rotation). These conformations are reminiscent of eEF2-containing TI-POST−1 and −2 states[85], which represent ribosomes that have not undergone full translocation. During translation, GTP hydrolysis occurs late in the elongation process and is only required for the resolution of late TI-POST states. There it facilitates dissociation of eEF2 and formation of the bona fide POST-state. As a result, the ribosome is bound to a fully translocated tRNA₂-mRNA module and the A site is empty. Interestingly, in our classes, eEF2 is bound to GDP,

rather than GTP, yet eEF2 is still present in the A site. Based on these results, we conclude that nelfinavir treatment induces formation of ribosomes containing eEF2 bound to GDP and SERBP1.

**Phosphorylation induces disorder in switch I of eEF2.** An important difference between the two classes resides at the eEF2K phosphorylation site, Thr56 (Figs. 5D, E and 6A, B). GTPases, including eEF2, possess conserved regions, termed switches, that

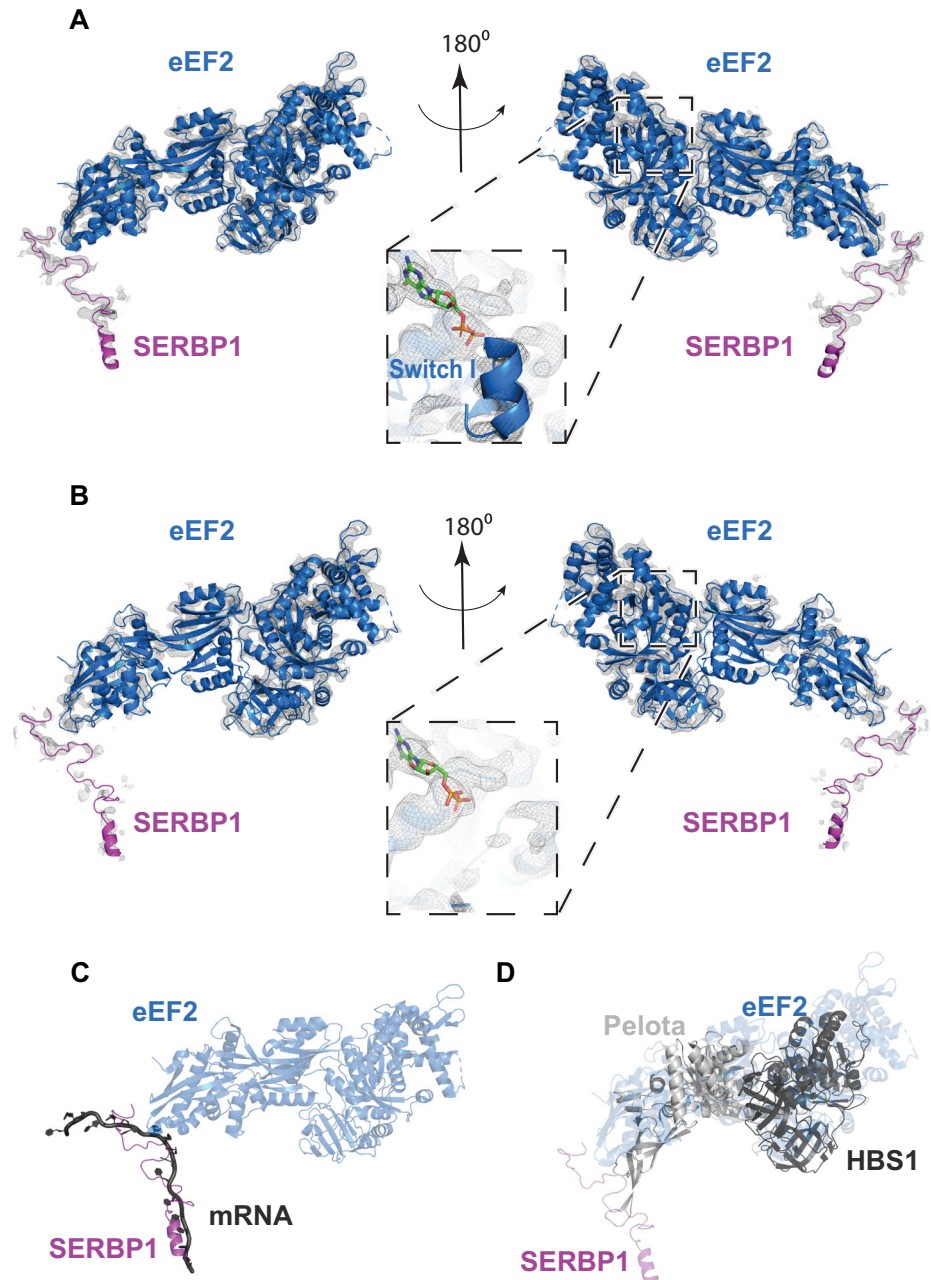

**Fig. 6 Density maps of switch I, and overlays of SERBP1, and eEF2 with mRNA, and recycling factors. A** View of eEF2 and SERBP1 of class I, and **B** class II. **C** For comparison on SERBP1 with a canonically bound mRNA, we aligned class I 28S rRNA to 28S rRNA from PDB ID 2Y0G. SERBP1 (purple) occupies the mRNA channel for the ribosome, thus excluding mRNA (dark gray) binding. **D** The overlay of class I with a recycling factor Hbs1/Pelota-bound ribosome (PDB ID 5LZX) illustrates that recycling factor Pelota and Hbs1-binding is mutually exclusive with bound eEF2/SERBP1. Class I 28S rRNA was aligned to 28S rRNA of the Hbs1/Pelota-bound ribosome.

are integral to their activity. Due to interactions with the GTP gamma-phosphate, switch I adopts an ordered conformation in the presence of GTP and becomes disordered after hydrolysis[85]. A transition state induced with a GDP•Pi analog and sordarin in which switch I contacts nearby rRNA of the 40S shoulder region has been reported as well[86]. This suggests that conformational dynamics are an integral part of eEF2 function.

Switch I (residues 53–72) harbors the eEF2K-dependent phosphorylation site, Thr56. Both switch I and switch II (residues 106–124) monitor the hydrolysis state of bound GTP[87]. Switch I has visible density in class I, yet, in class II the switch I region

appears to be disordered at comparable display thresholds (Figs. 5E and 6A, B). In the structured switch I, unphosphorylated Thr56 is oriented towards the bound GDP and is surrounded by negatively charged residues (Figs. 5D and 6A). This suggests that phosphorylation of Thr56 may cause disorder of switch I due to electrostatic repulsion. Elaborate image processing strategies including masking and signal subtraction were not successful at resolving the switch I region of class II suggesting that phosphorylation leads to conformational heterogeneity of switch I rather than a single alternative conformation, however precluding the GTP-sensing conformation. Thus, our data

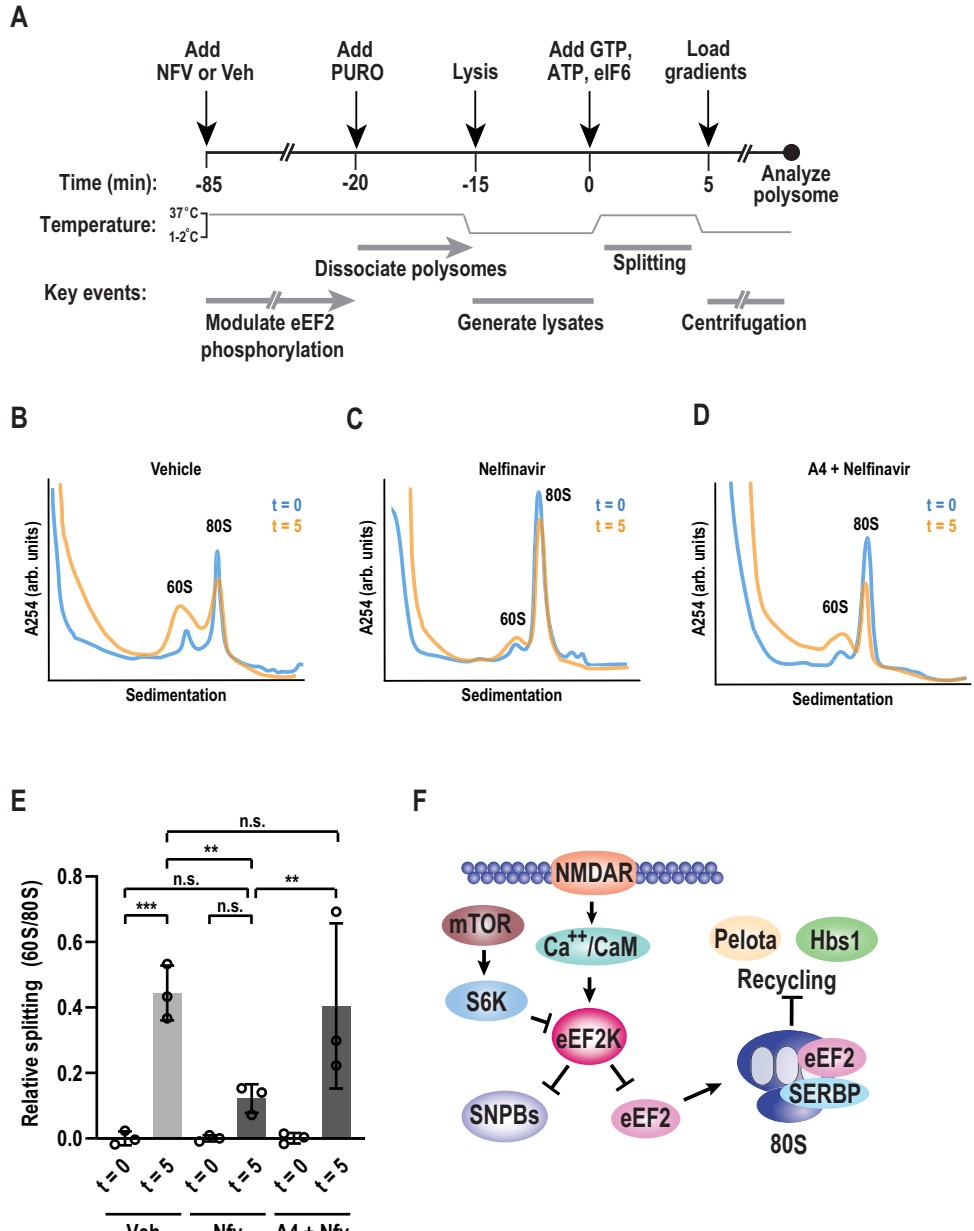

**Fig. 7 Nelfinavir-induced monosomes are resistant to recycling. A** Schematic diagram of in vitro splitting assay. Cells were treated for 1 h with vehicle (Veh) or nelfinavir (NFV), followed by 50 μM puromycin (PURO) for 5 min. Cells were then lysed and clarified by centrifugation. Splitting assays were initiated with the addition of ATP (1 mM), GTP (1 mM), and eIF6 (5 μM), and transferred to 37 °C for 5 min. Reactions were halted by cooling samples on ice before performing polysome profiles. **B** Representative polysome profiles from splitting assays following vehicle treatment performed pre-splitting (t = 0, blue) and post-splitting assay (t = 5, orange). **C** Representative polysome profiles from splitting assays following nelfinavir treatment performed pre-splitting (t = 0, blue) and post-splitting assay (t = 5, orange). **D** Representative polysome profiles from splitting assays in cells pre-treated with A4 (25 μM) followed by nelfinavir treatment performed pre-splitting (t = 0, blue) and post-splitting (t = 5, orange). **E** Quantification of relative splitting, as measured by the ratio of 60S peak height to 80S peak height. Initial ratios (t = 0) were subtracted from corresponding treatment groups. Error bars represent mean ± SD. n = 3 biological replicates. P-value determined by one-way ANOVA. Veh t = 0 vs Veh t = 5 p = 0.0004, Veh t = 0 vs NFV t = 5 p = 0.2039, NFV t = 0 vs. NFV t = 0 p = 0.2039 Veh t = 5 vs NFV t = 5 p = 0.0038, NFV t = 5 vs A4+NFV t = 5 p = 0.0086, Veh t = 5 vs. A4+NFV t = 5 p = 0.6673. **F** A proposed model highlighting eEF2K functions in sensory neurons. Source data are provided as a Source Data file.

suggest that phosphorylated eEF2 is capable of occupying the A-site of translationally inactive monosomes.

**eEF2K inhibits recycling of vacant 80S ribosomes**. A comparison with structures containing the mammalian recycling factors Pelota and Hbs1, which promote dissociation of stalled ribosomes, suggests that their association is mutually exclusive with

SERBP1 and eEF2 (Fig. 6D). We therefore hypothesized that nelfinavir impacts ribosome recycling. We adapted an in vitro splitting assay to interrogate this problem (Fig. 7A)[88]. F11 cells were treated with either vehicle or nelfinavir to modulate eEF2K activity. Afterward, polysomes were dissociated with puromycin. Cells were lysed and clarified by centrifugation. eIF6 was added to prevent reassociation of the 40S and 60S subunits[88–91]. Assays were initiated with the addition of GTP and ATP and conducted

at 37°C. Splitting reactions were then used to generate polysome profiles, and splitting efficiency was assessed based on the relative accumulation of 60S subunits. After 5 min, we found that the 60S/80S ratio was drastically increased in the vehicle treated samples, suggesting efficient splitting of subunits (Fig. 7B). To determine if splitting was mediated by Pelota/Hbs1, we conducted a control where Pelota was depleted using immunoaffinity precipitation. Comparison of Pelota depleted samples to a mock depleted sample revealed that splitting was significantly reduced (Fig. S6). Next, we examined samples treated with nelfinavir (Fig. 7C). Splitting was reduced by roughly 72% compared to the vehicle treated group (Fig. 7E). To validate that the effect on splitting was due to nelfinavir's enhancement of eEF2K activity, we repeated the assay on cells pre-treated with A484954. Inhibition of eEF2K prior to nelfinavir treatment resulted in substantial recovery of ribosome splitting (Fig. 7D, E). Based on these observations, we propose a model where activation of eEF2K promotes the stabilization of 80S ribosomes by preventing their recycling concurrent with p-body repression (Fig. 7F).

## Discussion

Our data enable four major conclusions. First, the generic relationship between mRNA association with polyribosomes and the abundance of p-bodies is fundamentally different in primary sensory neurons versus mitotically active cell lines. Second, the eEF2K agonist nelfinavir induced a near complete loss of SNPBs that was concurrent with repression of translation. Third, we found that nelfinavir induced eEF2-phosphorylation and led to stabilization of inactive 80S ribosomes. One of the structural classes together with biochemical experiments reveal that phosphorylated eEF2 associates with inactive ribosomes. Fourth and finally, we found that 80S ribosomes induced by nelfinavir were resistant to recycling.

The relationship between p-bodies and translation is distinct in sensory neurons. Experiments conducted in cell lines have led to a model that links p-bodies and translation via mRNAs that shuttle between ribosomes and p-bodies. This would predict that arresting translation by stabilizing vacant ribosomes would increase SNPB abundance. We found that stimulating eEF2K activity attenuates translation while simultaneously leading to a near loss of SNPBs. Additionally, dissociation of mRNA from translating ribosomes by puromycin failed to trigger a substantial increase in SNPB abundance. This is markedly different from both yeast and HeLa cells[22]. There are several potential explanations for this discrepancy. All of our experiments that examined SNPBs were conducted in primary and not immortalized cells. Additionally, neurons are terminally differentiated and do not undergo mitosis. Cell identity may also play a role in defining granule dynamics. Mice with abnormal eEF2K activity are overtly normal and fertile. Yet, they display abnormal learning and memory[92,93]. This suggests that eEF2K has tissue-specific functions that are particularly prominent in the nervous system. A potential caveat to our measurements is that we did not test a wide range of concentrations and timepoints. Nevertheless, our data suggest that mRNA is not rate-limiting component for SNPB formation and that the relationship between eEF2K activity and p-body abundance differs between cell types. The original characterization of neuronal p-bodies demonstrated cell-type specific differences in p-body constituents[17]. Our work extends this notion and suggests that p-body-like structures in different cell types may be governed by fundamentally distinct mechanisms.

We uncover a previously undescribed role for eEF2K in the regulation of protein synthesis. Our data establish that increased eEF2K activity stabilizes inactive 80S ribosomes that contain eEF2 in the acceptor site and SERBP1 in place of mRNA. How do they

form? Biochemical data indicate that phosphorylated eEF2 is present on these ribosomes. We did not observe vacant monosomes with SERBP1 in the mRNA channel in the absence of eEF2. The vast majority of cellular SERBP1 is bound to ribosomes[94]. SERBP1 also associates stably with 40S subunits, likely via a helix bound at the 40S eS10 and eS12 proteins. This implies that the presence of SERBP1 alone is not sufficient to inactivate ribosomes. However, our experiments are entirely consistent with a key role for SERBP1 in the stabilization of vacant ribosomes as it is known to conditionally insert itself into the mRNA channel. While the molecular mechanisms that trigger occlusion of the mRNA channel and possibly eviction of an mRNA by SERBP1 are unclear in mammals, it is conceivable that translational inhibition by SERBP1 promotes association of phosphorylated eEF2 with ribosomes[95,96]. To precisely define the order of these events, re-constitution experiments are necessary.

What regulates disassembly of vacant ribosomes? Based on starvation-induced 80S ribosomes found in S.cerevisiae, recycling may depend on prior eEF2 dissociation[28,29]. It is unclear what role the loss of phosphorylation on Thr56 plays in the dissociation of these 80S ribosomes. Our data suggest that vacant ribosomes are resistant to splitting but are eventually recycled in a Pelota-dependent mechanism. How this is regulated remains unclear. Dephosphorylation of eEF2 Thr56 might promote spontaneous dissociation of eEF2 and SERBP1. Removal or addition of post-translational modifications to SERBP1 may also play a role in regulating the stability of vacant ribosomes. Due to the absence of 80S ribosomes with either eEF2 or SERBP1 alone, we propose that eEF2 and SERBP1 cooperatively exclude 80S ribosomes from translation and prevent them from recycling. Given that a range of cues including energy deficiency and hypoxia stimulate eEF2K, temporary storage of ribosomes could be a common outcome of cellular stress.

Why is ribosome availability linked to SNPB abundance? A critical component to answering this question is first understanding the precise function of SNPBs. While they may store poorly translated mRNAs, their abundance is not broadly coupled to the availability of free mRNA. It is therefore unclear if mRNA storage is their primary role, consistent with prior work in yeast[97,98]. Yet, we can speculate as to how the SNPBs and translation might be mechanistically linked downstream of eEF2K. The most parsimonious explanation for eEF2K activation and repression of translation are effects on eEF2. eEF2 phosphorylation incapacitates its role in translation elongation. We propose that attenuation of translation also results from the generation of inactive ribosomes, which could serve to sequester eEF2 and limit ribosome availability. The relevant downstream target of eEF2K that affects SNPBs is less certain. For example, hyperactive eEF2K may trigger phosphorylation and inactivation of a factor that promotes SNPBs. Given that eEF2 is the sole known substrate of eEF2K, it is difficult to guess the identity of this factor. However, remarkably few kinases subject to intense scrutiny act on a single site in the cellular proteome. A second possibility is that SERBP1 and/or eEF2 is rate limiting for SNPBs and phosphorylation of eEF2 sequesters them on ribosomes. This mechanism would be surprising as, to the best of our knowledge, neither factor has been reported as a stable component of p-bodies. Yet, it may account for the reduction of SNPBs following treatment with emetine, as generating stalled eEF2-accessible polysomes may similarly sequester eEF2. A third possibility is that loss of SNPBs is an indirect consequence of stabilizing inactive 80S ribosomes. Numerous processes that are likely also impacted include: an increase in free mRNA, a decrease in free ribosome subunits, an increase in free initiation factors, an increase in recycling factors, and changes in the levels of charged tRNAs. An important question moving forward is resolving the

precise combination of mechanisms that link eEF2K and SNPBs. Given the key roles of eEF2K in stress and plasticity, deciphering this mechanism may reveal insights into the function and purpose of SNPBs.

In summary, we have uncovered unanticipated roles for a conserved elongation factor in the control of SNPBs. eEF2K regulates ribosome availability through the generation of vacant 80S particles that are resistant to recycling. This presents an intriguing scenario in which elongation factor regulation may directly modulate initiation via the sequestration of recycling-resistant 80S ribosomes. We suggest that the standard translation cycle (initiation, elongation, and termination/recycling) neglects a key aspect of translation. Notably, re-appropriation of elongation factors to form inactive ribosomes that resist recycling. This might have important consequences on the number of ribosomes avaiable for translation.

## Methods

**Animals**. All procedures that involved use of animals were approved by the Institutional Animal Care and Use Committee of The University of Texas at Dallas. Animals were housed at an ambient temperature of 22.2 °C and 50–58% humidity with a 12-h light/dark cycle. Food and water was available ad libitum. Swiss Webster (WT) mice (Mus musculus) were obtained from Taconic Laboratories. eEF2K KO mice were originally generated by Alexey Ryazanov[73]. A breeding pair was generously provided to us by Tao Ma.

**Primary DRG culture**. DRG tissues were extracted from male mice between four and five weeks of age. In brief, after the animal was sacrificed, the entire spine was removed and hemi-sectioned. The spinal cord and dura were removed from each hemi-section. Individual ganglia were gently picked from between each pair of vertebrae using fine forceps and placed in ice-cold HBSS (Thermo). Tissues were centrifuged for one minute at $400 \times g$. The HBSS was aspirated and the DRGs were resuspended in solution A (1 mg/ml collagenase A in HBSS) followed by incubation for 25 min at 37 °C. The tissue was then centrifuged for 1 min at $400 \times g$, the supernatant removed, and tissue resuspended in solution D (1 mg/ml collagenase D, 10% Papain in HBSS). Following incubation for 20 min at 37 °C, the tissue was centrifuged for an additional minute at $400 \times g$, supernatant removed, and tissue resuspended in solution T (1 mg/ml Trypsin inhibitor, 1 mg/ml BSA in TG media). The tissue was triturated until a homogenous mixture was formed, then pipetted over a 70 μM cell strainer, with the cells collected in a Falcon tube. To remove residual cells, the strainer was washed with 15–20 ml warm DMEM/F12. The cell suspension was centrifuged for 5 min at $400 \times g$. The media was removed, and the cells resuspended in DRG culture media to achieve a confluency of 60%. The culture media consists of DMEM/F12 + GlutaMAX, 10% FBS, 1% penicillin/streptomycin, 3ng/ml 5-Fluoro-2′-deoxyuridine, and 7 ng/ml uridine. After plating, media was replenished every other day. For ICC, DRG neurons from one male animal were used for each 8-well slide, with biological replicates performed independently from separate cultures. For immunoblots, DRG neurons from 4 male animals were pooled for each 6-well plate, with biological replicates performed in parallel. Swiss-Webster mice were used for all primary cultures except where indicated.

**U2-OS culture**. U2-OS (RRID CVCL-0042; ATCC) cells cultured in DMEM containing 10% FBS and 1% penicillin/streptomycin. For immunocytochemistry experiments, $1.8 \times 10^4$ cells were plated per well of an 8-well chamber slide (Lab-Tek). In the immunoblot experiments, cultures were seeded at a density of $3 \times 10^5$ cells per well of a six well (9.6 cm²) tissue culture plate (Corning). Cells were grown to approximately 70–80% confluency prior to use in assays.

**F11 culture**. F11 (ECACC 8062601; Sigma 08062601-1VL) cells cultured in DMEM containing 10% FBS and 1% penicillin/streptomycin. For polysome profiles and splitting assays (see below), $2.2 \times 10^6$ cells were plated on a 10 cm tissue culture dish (one per replicate). Cells were grown to approximately 70–80% confluency prior to use in assays.

**Immunocytochemistry**. DRGs were plated on 8-well chamber slides (LabTek) coated with poly-D-lysine and cultured for 5 days. After use in an assay, cultures were washed once with warm PBS then fixed for 15 min in 4% formaldehyde. Cultures were washed three times with wash buffer (1% BSA in PBS, same for all subsequent washes). Afterward, cells were permeabilized with 0.5% TritonX100 (in PBS) for 5 min. To remove the detergent, samples were washed three times. Samples were blocked with addition of 8% goat serum (Sigma, diluted in wash buffer) for 1 h at ambient temperature (22–24 °C). After blocking, the serum was aspirated and primary antibodies diluted in 8% goat serum were added onto the samples and allowed to incubate overnight at 4 °C. DRG neurons were labeled with

antibodies against RCK (MBL, 1:1000), peripherin (Novus, 1:1000), and eEF2K (Invitrogen, 1:500). U2-OS cultures were labeled with antibodies against RCK (1:500, SCBT), and phalloidin-TRITC (1:200). Samples were washed three times before adding secondary antibodies (all 1:1000) diluted in 8% goat serum. After for 1 h, samples were washed three times and nuclei were stained with DAPI (0.1 ng/ml, Sigma) for 10 min. The chambers were removed from the slides, and coverslips were mounted using ProLong Glass antifade mountant (ThermoFisher). Slides were fixed using clear nail polish.

**Fluorescent non-canonical amino acid tagging (FUNCAT)**. Samples were processed in the same manner as in ICC with the following modifications. Prior to treatments, cells were incubated in methionine-free media for 30 min. AHA (Click Chemistry Tools, 50 μM) was added for the last 30 min of treatment. Following permeabilization, cells were incubated in label mix (5 mM $CuSO_4$, 5 mM THPTA (Lumiprobe), 8 μM alkyne-conjugated sulfo-Cy5 (Lumiprobe), 4 mg/ml ascorbic acid in 50% DMSO) for 30 min followed by three washes with click wash buffer (1% Tween-20, 0.5 mM EDTA in PBS). Samples were then subjected to the remainder of the ICC protocol (above).

**Processing of single-cell sequencing data**. Harmony-corrected principle component analyses was performed in R on the GSE59739 dataset [76,99]. 5538 genes were excluded from PCA as they contain zero variance. The remaining 19,799 genes were used to generate the PCA plot. Moderate cluster separation was preserved over multiple combinations of principal components, although clusters never completely separated. Five distinct clusters, one non-neuronal and four neuronal, were identified and characterized according to validated marker genes[76]. Cluster identity was defined based on groups of cells that share expression of marker genes corresponding to a particular cell type. This was used to guide the placement of boundary regions on the t-SNE plot. Cells which localized within overlapping borders of known cell-type clusters were unable to be discretely categorized to a single cell-type. Two parallel quantifications were conducted; one counting percent localization of only cells with discrete cell-type clustering, and one including cells with imperfect clustering when counting percent localization by treating those cells as both cell-types they cluster into. Each quantification was considered the minimum or maximum percent co-expression, respectively, and used to determine the average eEF2K co-expression percentages.

**Immunoblots**. DRG neurons were cultured on poly-D-lysine coated 6-well tissue culture plates (Corning) for five days before treatment. Following treatment, cells were washed with ice-cold PBS, and lysed with RIPA buffer (150 mM NaCl, 0.1% TritonX100, 0.5% sodium deoxycholate, 0.1% SDS, 50 mM Tris-HCL, pH 8.0) supplemented with Pierce Protease Inhibitor and Pierce Phosphatase Inhibitor (Thermo). Lysates were centrifuged at 4 °C for 20 min at $18,400 \times g$ and the supernatant was collected. Protein concentration was determined via BCA assay. 10 μg of protein was loaded into each well of a 12% SDS-PAGE gel and run at 100 V until fully resolved. Proteins samples were then transferred from the acrylamide gel onto a methanol activated PVDF membrane (Millipore) for 1.3 h at 100 V. Afterward, the membranes were blocked with 5% milk in TBST for 1 h at ambient temperature, followed by overnight incubation in primary antibody diluted in blocking solution at 4 °C (1 h at room temperature for GAPDH). DRG and U2-OS samples were blotted for p-eEF2 (CST, 1:1000), eEF2 (CST, 1:1000), and GAPDH (Proteintech, 1:10,000). DRG and F11 ribosome isolations were additionally probed with antibodies against SERBP1 (Bethyl, 1:1000), RPL5 (Bethyl, 1:1000), RPS6 (CST, 1:1000), and ATF-4 (CST, 1:1000). Blots were washed in TBST then incubated for 1 h at room temp in secondary antibodies conjugated to HRP (1:10,000). Immobilon® ECL Ultra Western HRP Substrate (Millipore) was added to the surface of the membrane for 2–4 min before visualizing. Band intensity was measured with Image Lab 6.0 (BioRad). Uncropped blot images are provided in the Supplemental Information (Fig. S7).

**P-body quantification**. Imaging was conducted using an Olympus FV3000 Laser Scanning confocal microscope on a 100X objective. Z projection of all images was performed with FluoView (Olympus) software. P-bodies were quantified for individual cells in Fiji[100] as follows. A region of interest (ROI) was manually drawn around the soma of a peripherin positive cell. Background signal subtracted using a rolling ball radius of 3. A threshold was applied before the image was converted to a mask. The Analyze Particles tool was then used to count RCK-positive puncta larger than 0.1 μm² with circularity greater than 0.6. This was repeated for 14–20 cells per condition. One-way ANOVA with multiple comparisons was used to compare the mean of each treatment group to the relevant control.

**Corrected total cell fluorescence (CTCF)**. Images were collected an Olympus FV3000 Laser Scanning confocal microscope through a 20X objective. Z projection of all images was performed with Olympus FluoView software. Fluorescence intensity was quantified for individual cells in ImageJ as follows. A region of interest (ROI) was manually drawn around the soma of a peripherin positive cell. In the Cy-5 channel, the Integrated Density (ID) of this ROI was measured. The background ID for each image was measured as the average of five background ROIs. CTCF for each cell was calculated as: cell ID − (background ID x cell area).

This was repeated for 25–30 cells per condition. All measurements for each experiment were then normalized by subtracting the average CTCF value of the no AHA group. Normalized CTCF values are expressed as a fraction of the vehicle treated average CTCF. One-way ANOVA with multiple comparisons was used to compare the mean of each treatment group to the vehicle treated control.

**Colocalization**. Images were collected with an Olympus FV3000 Laser Scanning confocal microscope on a 100X objective. Colocalization of eEF2K with RCK immunofluorescence was quantified in Fiji. ROIs were manually drawn around peripherin-positive cell bodies. To ensure colocalization was measured with genuine SNPBs, a threshold was applied to eliminate diffuse Rck signal. To quantify eEF2K colocalization with RCK puncta, Pearson's correlation coefficient was measured using the Coloc 2 tool.

**Polysome profiles**. Prior to lysis, cells were treated with 100 μg/ml cycloheximide (except for splitting assays, see below) for 5 minutes. Cells were washed in ice-cold PBS (supplemented with 100 μg/ml cycloheximide), lysed in polysome lysis buffer B (20 mM Tris-HCl pH 7.5, 150 mM NaCl, 5 mM MgCl$_2$, 1 mM DTT, 40 U/ml RNasin Plus Rnase Inhibitor, Dnase I, Pierce Protease and Phosphatase inhibitors, 100 μg/ml cycloheximide), and crude lysates were centrifuged for 10 min at 13,000 x $g$ to pellet debris. Clarified lysates were layered on 10–50% sucrose gradients (prepared in 20 mM Tris-HCl pH 7.5, 150 mM NaCl, 5 mM MgCl$_2$) and centrifuged for 2 h at 190,000 × $g$. Gradients were fractionated using an NE-1000 syringe pump (New Era Pump Systems, Inc.) and 254 nm absorbance was measured using an ISCO Type 11 optical unit and UA-6 detector.

**Ribosome purification by sucrose cushion**. $2.2 \times 10^6$ F11 cells were plated per 10 cm plate and treatments were conducted the following day after cells had achieved 70–80% confluency (For primary DRG neurons, cells from 6 animals were plated per poly-D-lysine coated 10 cm plate and cultured for 6 days). Cells were treated with vehicle (DMSO) or nelfinavir for 1 h, followed by 100 μM emetine for 5 min. Cells were washed with ice-cold PBS (supplemented with 100 μM emetine), lysed with polysome lysis buffer A (25 mM Hepes-KOH, pH 7.2, 110 mM KOAc, 2.5 mM Mg(OAc)$_2$, 1 mM EGTA, 1 mM DTT, DNase I, 40 U/ml RNasin Plus RNase Inhibitor (Promega), 0.015% digitonin, supplemented with Pierce Protease Inhibitor and Pierce Phosphatase Inhibitor (Thermo) and 100 μM emetine), and removed from the plate with a cell scraper. Crude lysates were collected and centrifuged at 13,000 × $g$ for 10 min at 4 °C to remove debris. The clarified lysate was then loaded onto 0.5 ml 30% sucrose cushion (20 mM Tris pH 7.5, 2 mM Mg(OAc)$_2$, 150 mM KCl, 30% w/v sucrose, supplemented with RNaseIN Plus RNase inhibitor (Promega)). Ribosomes were pelleted by ultracentrifugation at 120,000 × $g$ for 24 h at 4 °C using a Beckman Coulter S55A fixed-angle rotor. Pellets were resuspended in polysome lysis buffer.

**Rapid ribosome isolation by size exclusion chromatography (SEC)**. Our ribosome isolation method was adapted from Behrmann et al.[84] Briefly, primary DRG neurons were cultured on 10 cm cell culture plates coated with poly-D-lysine for 6 days. Following treatment, cells were washed with ice-cold PBS, lysed with polysome lysis buffer A (25 mM Hepes-KOH, pH 7.2, 110 mM KOAc, 2.5 mM Mg(OAc)$_2$, 1 mM EGTA, 1 mM DTT, 40 U/ml RNasin Plus RNase Inhibitor (Promega), DNase I, 0.015% digitonin, 100 μM emetine) supplemented with Pierce Protease Inhibitor and Pierce Phosphatase Inhibitor (Thermo), and removed from the plate with a cell scraper. Lysates were collected and centrifuged at 500 × $g$ for 10 min at 4 °C to remove cell debris. S400 Sephacryl spin columns (GE) were washed 6 times with equilibration buffer (20 mM Hepes-KOH, pH 7.5, 100 mM KCl, 1.5 mM MgCl$_2$, 0.5 mM spermidine, 0.04 mM spermine, 1 mM DTT). Lysates were then immediately loaded onto columns and spun for 3 min at 600 × $g$ at 4 °C to collect the heavy fraction (fraction 1, used for cryoEM). To collect the light fraction (fraction 2), additional polysome lysis buffer A was added to the columns, which were again centrifuged for 3 min at 600 × $g$.

**Cryo-EM specimen preparation**. C-flat grids (Copper, 300 mesh, 1/2, Protochips) were glow-discharged for 30 s at 15 mA in a PELCO glow-discharge unit. We estimated the input using A260 measurements. We applied 3 μl of the purified ribosomes with an absorbance at 260 nm of 7.5 to the grid. We incubated the sample for 30 s at 4 °C and >90% humidity, blotted for 3 s using blot force 3, and vitrified the sample in liquid ethane using a Vitrobot Mark IV (ThermoFisher).

**Cryo-EM data collection**. The data was collected in two sessions on a Titan Krios operating at 300 kV and equipped with a K3 camera (Gatan) and an energy filter. We automated data-collection using SerialEM[101]. To overcome preferential orientation of the sample, we tilted the stage to 35°. We calibrated coma vs. image shift and collected 2–3 images per hole using the multi-shot option implemented in SerialEM. The dataset is comprised of 3583 movies collected in super-resolution mode and saved dark-corrected. The calibrated pixel-size is 0.53 Å in super-resolution mode. Nominal defoci ranged between −0.5 to −2.5 μm. Each movie comprised 75 frames with a total dose of 75 e-/Å$^2$.

**Image processing**. All image processing was done using cisTEM[102]. Dark references were calculated as described before by Afanasyev et al.[103] and used to correct the movies. The movie frames were aligned using unblur within cisTEM. CTF-parameters, tilt angle and axis were estimated in an updated version of CTFfind4[104], which is implemented within the latest development version of cisTEM (available on github: [https://github.com/ngrigorieff/cisTEM])[102]. Images with ice contamination or poor CTF fits were excluded from further processing, yielding a dataset of 2995 movies from which we picked 193,794 coordinates using the "find particles" function. We then extracted the particles in 768 pix$^2$ boxes.

We generated an ab initio model from 25% of the data. The reconstruction was further refined using the "auto-refinement" function with auto-masking disabled. Next, we ran a global search aligning all particles to 30 Å to the ab initio model followed by 10 rounds of refinement with increasing resolution limits to 5 Å. The final reconstruction was subjected to CTF-refinement to 3.5 Å without coordinate or angular refinement. The resulting reconstruction reached a resolution of 3.0 Å and showed eEF2-density in the A site.

Classification with a focus mask around the A site (coordinates 400, 500, 390, and radius 60 Å) into six classes yielded three classes with eEF2 density in the A site and tRNA in the E site, one class representing large subunits and damaged particles, and two classes without eEF2 (for detailed classification scheme see Fig. S4A). We then merged all eEF2-containing classes and aligned them to a common reference to 5 Å. Finally, we classified without alignment with a focus mask around domains I and II of eEF2 (coordinates 410, 490, 475, and radius 35 Å). The obtained classes reached resolutions between 3.1 and 3.3 Å. Two classes contained density corresponding to eEF2 in the A site.

**Model building**. For model building, we generated several sharpened maps with B-factors from −30 Å$^2$ to −90 Å$^2$ using cisTEM, and in Phenix.autosharpen[105]. The initial model was obtained by fitting the large subunit, small subunit head, and small subunit body of a human ribosome (PDBID 6ek0) individually into the density using Chimera. To generate an initial model for the mouse ribosome we changed residues manually in Coot[106,107] and inspected the map closely for conformational differences. Next, we fit rabbit eEF2 (PDBID: 6mtd) and changed residues to match the murine eEF2 sequence (UniProtKB P58252). We refined the model in Phenix using phenix.real_space.refine and manually corrected outliers in Coot. The resulting models were evaluated in MolProbity[108].

**Molecular cloning**. The eIF6 insert was amplified from mouse cDNA using the following primers: 5′-CATCCTCCAAAATCGGATCTGGTTCCGCGTGGATCC CCGGAATTCATGGCGGTCA GAGCG −3′ (5′ primer) and 5′-TCACCGAAA CGCGCGAGGCAGATCGTCAGTCAGTCA ACGATGCGGCCGCTGTGAGGCT GTCAATGAGG-3′ (3′ primer). pGEX-4T:eIF6 was generated by Gibson assembly[109]. eIF6 insert and pGEX-4T linearized with NotI (Thermo) and EcoRI (Thermo) were added to Gibson assembly mix at a 6:1 molar ratio and incubated for 1 h at 50 °C. The Gibson product was then used to transform competent DH5α, which were then plated on LB + ampicillin plates and incubated overnight at 37 °C. Single colonies were used for overnight cultures. pGEX-4T:eIF6 was purified from overnight cultures using GeneJET Plasmid Miniprep Kit (Thermo) and validated by Sangar sequencing. This vector can be obtained from the corresponding author on request.

**Protein purification**. Starter cultures of BL21 codon plus transformed with pGEX-4T:eIF6 were grown overnight at 37 °C in LB media supplemented with ampicillin and chloramphenicol. The starter cultures (5 ml) were used to inoculate 1 liter of media supplemented with ampicillin and chloramphenicol. The large-scale cultures were grown at 37 °C with shaking at 225 rpm for 4.5 h then shifted to at 15 °C at 200 RPM for 1.5 h. Protein expression was induced with 0.5 mM IPTG for 16 h at 15 °C with shaking at 200 rpm. The large-scale cultures were centrifuged at 7500 × $g$ for 45 min. The bacterial pellets were resuspended in 35 ml of Resuspension Buffer (50 mM Tris-HCl pH 8.0, 500 mM NaCl, 5 mM DTT, 0.2% NP-40, 5% glycerol, 1 mM EDTA, 20 mM BME, 1 mg/ml lysozyme, 1 mM PMSF, Pierce Protease Inhibitor (Thermo)). The bacterial suspensions were sonicated at the following settings: Power 70%, on/off cycle for 3 s, each for 2 min twice. The lysate was centrifuged at 28,000 × $g$ for 20 min at 4 °C. The supernatant was loaded onto 2 ml of pre-equilibrated GST agarose resin in polypropylene chromatography columns and allowed to flowthrough under gravity. The loaded columns were washed with 100 ml of wash buffer (50 mM Tris-HCl pH 8.0, 1 M NaCl, 5 mM DTT, 5% glycerol). The bound protein was incubated with 4 ml of elution buffer (50 mM Tris-HCl pH 8.0, 300 mM NaCl, 5 mM DTT, 30 mM reduced glutathione, 5% glycerol) for 10 min at 4 °C and eluted. A second elution was performed to completely elute the bound protein. The eluted protein was dialyzed overnight at 4 °C at low speed stirring using snakeskin dialysis tubing in 2 liters of dialysis buffer (20 mM Tris-HCl pH 8.0, 300 mM NaCl, 0.1 mM PMSF, 5% glycerol). The dialyzed protein was concentrated using concentrator columns. BCA was used to estimate protein concentration.

**In vitro splitting assay**. F11 cells were grown to 70% confluency prior to treatment. Cultures were treated with 50 μM puromycin for 5 min following 60 min nelfinavir or vehicle treatment. Cell were washed with ice cold PBS and removed

from the plate with a cell scraper in 200 µl splitting buffer (20 mM Tris pH7.5, 100 mM KOAc, 4 mM Mg(Oac)$_2$, 1 mM DTT, 40 U/ml Rnasin Plus Rnase Inhibitor (Promega), Dnase I, and Pierce Protease and Phosphatase inhibitors. Cell suspensions were incubated on ice for 5 min before being lysed by passage through a 30 g needle. Lysates were cleared of debris by centrifuging at $13,000 \times g$ for 10 min at 4 °C. To prevent reassembly of 80S ribosomes, eIF6 (5 µM) was added to each reaction mix immediately before initiating the splitting assay[88–91]. ATP and GTP were added to a final concentration of 1 mM each before incubation at 37 °C for 5 min to allow splitting of 80s ribosomes. Samples were put back on ice before being layered onto sucrose gradients for polysome profiling (see above, 10–50% sucrose gradients for splitting assays were made in buffer containing 20 mM Tris pH7.5, 100 mM KOAc, 4 mM Mg(Oac)$_2$).

**Immunoprecipitation**. Pierce protein A/G magnetic beads (Thermo) were washed three times with tris-buffered saline. Beads were then mixed with rabbit anti-Pelota at a ratio of 12 µg antibody to 30 µl beads. The bead/antibody mixture was incubated overnight at 4 °C with end-over-end mixing. Beads were then washed three times with splitting buffer. To deplete lysates for splitting assays (above), 10 µl of the bead/antibody mixture was added per 200 µl of lysate and incubated on ice for 30 min (mock IP was performed with unbound beads). The depleted (or mock-depleted) lysate was separated from the magnetic beads and used for splitting reactions as described above.

**Reporting summary**. Further information on research design is available in the Nature Research Reporting Summary linked to this article.

## Data availability

The structural models and cryo-EM maps have been deposited in the PDB and EMDB under accession codes 7LS2 (class I) and 7LS1 (class II), and EMD-23501 (class I) and EMD-23500 (class II) [], respectively. Single-cell RNA-seq data used in this study were generated by Usoskin et al. are accessible through the Gene Expression Omnibus, under accession code GSE59739[75,76]. Source data for Figs. 1–4, 7, S1, S2, and S6 are provided with this paper. Uncropped blot images are provided in the supplement (Fig. S7). Source data are provided with this paper.

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

## Acknowledgements
We thank Niko Grigorieff and Tim Grant for making the development versions of
CTFFIND4 and cisTEM available to us, Aaron Goldstrohm for his kind suggestions, and
Alexey Ryazanov and Tao Ma for generating and providing the eEF2K mutant mice,
respectively. This work was supported by NIH grants R01NS100788 (ZTC) and
R01NS114018 (ZTC).

## Author contributions
S.L., Z.T.C. and P.R.S. conceptualized and designed this study. S.L. performed cryo-EM,
related structural analysis, and model building. A.D.S. performed computational analysis
of sequencing data. P.R.S. performed the remaining experiments with assistance from
N.K. and T.L. S.L., Z.T.C. and P.R.S. drafted, edited, and approved the final manuscript.
All authors have read and agreed to the published version of this manuscript.

## Competing interests
The authors declare no competing interests.

## Additional information
**Supplementary information** The online version contains supplementary material
available at https://doi.org/10.1038/s41467-021-27160-4.

**Peer review information**: *Nature Communications* thanks Pavel Ivanov and the other,
anonymous, reviewer(s) for their contribution to the peer review of this work. Peer
reviewer reports are available.

**Publisher's note** Springer Nature remains neutral with regard to jurisdictional claims in
published maps and institutional affiliations.

