## [Peer Review File · Nature Communications]

REVIEWER COMMENTS

Reviewer #1 (Remarks to the Author):

The function of P-bodies is an actively researched topic. Over the last years we saw a lot of progress, and the general perspective has changed, e.g. "PBs as sites of mRNA degradation and/or storage should be re-evaluated (Tutucci et al. Nat Methods 2018 Jan;15(1):81-89)". The current study is in line with this perspective (... maybe worth citing Tutucci et al.).

In this study Smyth and colleagues investigate the connection between P body assembly and translational shutoff mediated through eEF2 phosphorylation and recruitment of SERBP1/Stm1p. The experimental system is primary DRG neurons (in most of the experiments) or F11 cell line (cell hybrid between rat DRG and mice neuroblastoma), which is important when comparing the results with a body of literature generated in other experimental systems, both human and yeast. Translation (and its regulation) in neurons has various distinct features that set it apart from that in somatic cells (or yeast). The study uses an array of methods: optical microscopy (very, very nicely done!), molecular biology (varied degrees of elegance) and cryo-EM (some additional analyses would be appreciated, see below). I feel that the paper could be improved if some gaps would be closed through additional experimentation to improve the cohesiveness.

p. 5-6, section 'Translation inhibitors have inconsistent effects on SNPB abundance'. In the experiments presented in this section the authors use a single concentration of antibiotic (no titration) and use two experimental readouts – microscopy or FUNCAT reporting active translation through proteome via incorporation of a non-canonical amino acid. I am not sure that one can arrive at such a specific and strong conclusion as 'we reasoned that factors involved in translation elongation might play critical roles in coordinate regulation of translation and SNPBs': here we conflate antibiotics mode of action vs dose-dependent effects.

All of the treatments do result in decreased SNPB numbers and decreased translation. I would relegate these results to SI and start with eEF2K KO / Nelfinavir results directly and introduce Nelfinavir already in the introduction. We really do not learn much from the antibiotics study – and one wants to make it conclusive, one would need i) titrations ii) time courses.

(Note that the off-target effect of Nelfinavir that results in eEF2 phosphorylation is mediated not via eEF2K activation but via inhibition of eEF2-P dephosphorylation. This is never spelled out - and I am also not sure one can call Nelfinavir an eEF2K agonist / activator: it is not actually affecting eEF2K directly. It is an antagonist of CReP phosphatase (I would mention that in the introduction, seems like a key concept to be aware of...). Also I feel 'Pharmacological activation of eEF2K' section title is misleading: Nelfinavir does not activate eEF2K, the effect is mediated by unchanged basal activity of eEF2K coupled with CReP inhibition. ...OK, back to the data.)

The authors convincingly show that Nelfinavir can inhibit translation (as expected – eEF2-P is not translationally active) and decrease SNPB numbers dramatically. Here it is crucial to specify already in the Results section the concentrations used in the experiments and relate to the concentrations that are pharmacologically relevant: Nelfinavir is a drug. Are we looking at clinically relevant concentrations? 10x? 100x? 10000x? This should be clear for the reader. Note that in this section the authors use an eEF2K KO cell line as a specificity control. This strain will be very useful for some additional experiments (see below).

Then the authors show that the effects are different in somatic cells – and proceed to counteract the Nelfinavir effects by NMDAR activation by MK801. I feel that the intro should be covering eEF2K / CReP vs Nelfinavir / MK801 vs NMDAR. Maybe the section 'Rescue of SNPB loss by nelfinavir by an NMDAR antagonist' can be combined with 'Pharmacological activation of eEF2K causes loss of SNPBs and translational Repression': it is, more or less, a control experiment for Nelfinavir treatment

experiments? There is no new conceptual insight, all it shows is that yes, eEF2-P causes loss of SNPBs and translational Repression – and the authors have a pharmacological handle on the system. I would discuss the MK801 experiment before going for somatic cells (different system, new conclusion).

In the following sections the authors focused on the mechanistic aspects of the system. The question they ask is: is the translational shutoff by Nelfinavir resulting in polysome collapse and 80S stabilisation?

1. Polysome sucrose gradient fractionation experiments, Figure 5A. Here the authors stabilise polysomes with elongation inhibitor cycloheximide.
2. Ribosomal purification and immunoblotting, Figure 5B. Here elongation inhibitor emetine is used.

Why the two different inhibitors used? What is no antibiotic (cycloheximide or emetine) is used? Is eEF2 stabilised on the 80S by the antibiotics combined with phosphorylation – or is phosphorylation alone sufficient? Note that elongation inhibitors are used to stabilise polysomes. Here we have very little polysomes in the case of Nelfinavir-treated cells, therefore it is essential to perform an experiment without cycloheximide / emetine co-treatment as well, otherwise one can not make a decisive conclusion.

Why two using different approaches? The standard experimental approach here is 1) run sucrose gradients 2) collect fractions 3) do immunoblotting on these fractions 4) combine the immunoblotting data and the polysome data into one figure. In the current version the two assays are decoupled – and are performed differently. Is it the effect of emetine vs cycloheximide? What happens when there is no cycloheximide / emetine co-treatment? EDTA vs no EDTA? Finally, a nice control to have would be a sucrose gradient experiment (just the UV trace) using eEF2K KO cell line treated by Nelfinavir.

The next step is cryo-EM analysis of stable 80S formed upon Nelfinavir treatment. Additional analyses are needed:

- i) a figure showing local resolution of cryo-EM maps for classes of interest
- ii) zoom-in on the model and density for SERBP1
- iii) direct comparisons with other structures of translationally inactive mammalian ribosomes (e.g. <https://www.ncbi.nlm.nih.gov/pmc/articles/PMC6226290/>).

The focus of the figure is on eEF2-P – but it is possible that the SERBP1 who is doing the work of 80S stabilisation (see below), therefore it is essential to document how well resolved it is.

Next the authors ask a question of eEF2-P and SERBP1 protect the ribosomes from splitting. For this they use cells purified from Nelfinavir-treated cells. As they show in Figure 5, Nelfinavir treatment results in SERBP1 recruitment to 80S (and stabilises them?). Therefore, one can not make a decisive conclusion that it is eEF2-P that stabilises the ribosomes – it could be SERBP1, which would then promote the eEF2-P association (see <https://www.ncbi.nlm.nih.gov/pmc/articles/PMC6226290/>). Therefore, current assays do not necessarily answer the question 'We therefore asked if phosphorylated eEF2 protects ribosomes from recycling', it is also possible that instead they ask the question is translational shutoff resulting in SERBP1-mediated 80S stabilization stabilising the 80S?

At least in yeast i) SERBP1/Stm1p actively inhibits translation and as a consequence of that eEF2-unrelated elongation factor 3, eEF3, is stabilised on the 80S <https://www.ncbi.nlm.nih.gov/pmc/articles/PMC2764444/> and 2) deleting Stm1p-encoding gene makes translation more active (at least in lysates) <https://pubmed.ncbi.nlm.nih.gov/30619132/> . Therefore, my money is on SERBP1 being the driver here - not eEF2-P.

To ask the question the authors ask one should 1) purify 80S from Nelfinavir-untreated cells 2) purify eEF2-P and eEF2 3) purify SERBP1 4) perform splitting experiments with this reconstituted system.

Discussion: I suggest the authors carefully re-read the discussion section after the additional experiments – with a focus on causality links they infer from the data. The ideas they put forward are very interesting and will be important for the field: titration of eEF2 on inactive ribosome pool, interplay between ribosomal shutoff and active factor concentrations.

Reviewer #2 (Remarks to the Author):

The manuscript by Smith and colleagues describes how nelfinivir, a small molecule drug originally identified as an HIV protease inhibitor but also shown to be an eEF2K agonist, acts to reduce translation in primary sensory neuron culture. The authors show that nelfinivir dramatically reduces the number of sensory neuron processing bodies, and that this effect requires eEF2K. They also show that eEF2 phosphorylation inhibits translation in these cells by locking the ribosome into an 80S complex that contains phosphorylated eEF2, SERBP1, lacks mRNA, and is resistant to recycling by Dom34/HBS. Several lines of experiment, from imaging to polysome analysis to cryo-EM structure determination, were used to develop this model. The results are fascinating and of broad interest. I do have some comments for the authors to consider to improve the presentation. I recommend that the study be published following a minor revision to consider these suggestions.

1. The abstract should make it clear that nelfinivir's effect on p-body abundance is restricted to sensory neurons, as the data shows it has no effect in other cell types (U2-OS).
2. The last paragraph of the introduction is hard to follow and largely redundant with the results section. I suggest revising this paragraph to briefly summarize the take home messages rather than to foreshadow the experiments.
3. A figure similar to figure 6E would be useful earlier in the manuscript. It would be especially useful if it included the identity of the key reagents used in the study (nelfinivir, A484954, MK801, etc) and their modes of action.
5. It would be helpful to present the mode of MK801 inhibition prior to describing the data. It would help the audience to understand the rationale behind the experiment.
6. What are the implications of the non-rotated conformation and the comparison to rabbit reticulocyte lysate structure described on the bottom of page 10?

Minor comments:

1. In figure 2A/B, does A484954 lead to a significant increase relative to vehicle? If so, this should be addressed in the text.
2. In Fig4 panel B please add NFV only and MK801 only treated blots for comparison.
3. Fig5 panel B. I presume RPL5 and RPS6 are used as large subunit / small subunit loading controls. This should be clarified for the non-experts in the methods section.
4. There is a misspelling in FigS2 legend "FUCAT".
5. What was the rationale for the use of the U2-OS cell line in Fig S2 (as opposed to other non-neuronal lines)?

Reviewer #3 (Remarks to the Author):

The work by Smith et al. is dedicated to study roles of eEF2K, eEF2 in the control of ribosome availability and its relation to formation/abundance in sensory. Although the work is potentially interesting, I cannot support its publication in the current form at such prestigious journal as Nature Communications.

First of all the role of PBs generally is not well understood in any system. So by choosing studying them, authors should have a physiological relevance especially in their model, sensory neurons. Although they provide a general model as a part of figure 6E, I do see how PBs in general and then recycling part fits to the obtained results.

The work is fragmented and has no solid molecular mechanism data connecting two possible phenomena : formation of PBs and regulation of ribosome availability via eEF2K-ph-eIF2-SERBP1 axis.

The cryo-em part is solid but has been published earlier in other systems by other laboratories. While structural data is convincing, it does not provide any type of real quantification in vivo or in cells/neurons. What is the proportion of the ribosomes that become associated with SERBP1 upon nelfinavir treatment? I doubt that the assay based on the relative splitting can result in the adequate and unbiased quantification of such events.

The manipulation with KO cells are interesting but need rescue experiments with WT or even better rescue experiments with eIF2K mutants.

It is not clear why authors used only one single PB marker, RCK. In fact this marker is also SG marker. Moreover, the quality of IF pictures both for PB quantification and AHA incorporation is very low and needs better resolution.

The use of chemical inhibitors here such as MK801 that has several other pleiotropic effects besides NMDAR-related effects is unreliable.

It is also not clear why authors continued to study coupling between translation and SNBPs, if they conclude (Fig 1, and page 6) that "the ongoing protein synthesis and SNBP abundance are not directly coupled. Based on these result, we reasoned that factors involved in translation elongation might play critical roles in coordinate regulation of translation and SNBPs". This is puzzling!

Another aspect of the study that is critical is that authors do not show any dose- and time-response for any given experiment. How the concentration and time of treatment is chosen is not explained.

Reviewer #1:

“The function of P-bodies is an actively researched topic. Over the last years we saw a lot of progress, and the general perspective has changed, e.g. “PBs as sites of mRNA degradation and/or storage should be re-evaluated (Tutucci et al. Nat Methods 2018 Jan;15(1):81-89)”. The current study is in line with this perspective...”

We now cite the study as suggested by the reviewer.

“...All of the treatments do result in decreased SNPB numbers and decreased translation. I would relegate these results to SI and start with eEF2K KO / Nelfinavir results directly and introduce Nelfinavir already in the introduction....”

The changes in p-body abundance with Puro and HHT are not significant. Cycloheximide appears to be trending downward, but the apparent difference is also not significant. We have added N.S. labels to the figure to make this clearer to the reader.

While we appreciate the second concern regarding order of importance, it is our view that the simple fact that identical treatments have profound effects in cell lines is important. In the nociceptor field, the linkage between translation and p-bodies has been assumed to parallel that of cell lines (Paige C et al Neurobiol Pain 2019, Melemedjian OK et al Neurosci Lett. 2014 etc.). This stands in contrast to our study. To underscore the differences between systems, we have included an additional experiment where we repeated our drug treatments in U2OS cells using identical treatments to what we used in sensory neurons (Fig. 1B). Our data indicate that the concentrations and treatment times we used for sensory neurons yield the anticipated effect on p-bodies in a cell line after an identical amount of time as the prior experiment conducted with sensory neurons. These observations in conjunction with what we observe with Nelfinavir make a strong point that the mechanisms that govern SNPB number are distinct from their somatic counterparts.

“Note that the off-target effect of Nelfinavir that results in eEF2 phosphorylation is mediated not via eEF2K activation but via inhibition of eEF2-P dephosphorylation. This is never spelled out ... I feel ‘Pharmacological activation of eEF2K’ section title is misleading: Nelfinavir does not activate eEF2K, the effect is mediated by unchanged basal activity of eEF2K coupled with CReP inhibition...”

It would be ideal to include additional information about the mechanism of nelfinavir. However, our understanding of the precise mechanism of action is incomplete. CReP is responsible for dephosphorylation of eIF2a (Rojas M et al PNAS 2015). CReP partners with PP1. We were unable to find a link between nelfinavir and CREP (although nelfinavir may weakly influence PP1 activity - Gupta AK Neoplasia 2007).

To reframe the reviewer’s question ever so slightly - are the effects of Nelfinavir on eEF2 phosphorylation due to inhibition of a different phosphatase? eEF2 dephosphorylation occurs via PP2A (McDermott M Sj Mol Cancer 2014, Gergs U et al JBC 2004). However, nelfinavir does not impact PP2A activity (Ben-Romano R Diabetologia 2004). Thus, the mechanism that links Nelfinavir to eEF2K activation is unclear – although it is AMPK and mTOR independent (De Gassart A EMBO Rep. 2016).

“The authors convincingly show that Nelfinavir can inhibit translation (as expected – eEF2-P is not translationally active) and decrease SNPB numbers dramatically. Here it is crucial to specify already in the Results section the concentrations used in the experiments and relate to the concentrations that are pharmacologically relevant...”

We appreciate the suggestion - we are working at much higher concentrations (50 μ M) that what is achieved in patients (5.2 μ M). We specify the former concentration as suggested. We do not know if the effects we report would occur at a tenfold lower concentration. We are keen to follow up on this work with pharmacology experiments that assay pain associated behaviors in vivo, but these experiments are beyond the scope of the present work.

“Then the authors show that the effects are different in somatic cells – and proceed to counteract the Nelfinavir effects by NMDAR activation by MK801. I feel that the intro should be covering eEF2K / CReP vs Nelfinavir / MK801 vs NMDAR...”

We have added additional prose to the introduction that encompasses NMDARs but have not discussed CReP for reasons described above in the second point.

“Polysome sucrose gradient fractionation experiments, Figure 5A. Here the authors stabilize polysomes with elongation inhibitor cycloheximide. Ribosomal purification and immunoblotting, Figure 5B. Here elongation inhibitor emetine is used. Why the two different inhibitors used? What is no antibiotic (cycloheximide or emetine) is used? Is eEF2 stabilized on the 80S by the antibiotics combined with phosphorylation – or is phosphorylation alone sufficient?” Note that elongation inhibitors are used to stabilize polysomes. Here we have very little polysomes in the case of Nelfinavir-treated cells, therefore it is essential to perform an experiment without cycloheximide / emetine co-treatment as well, otherwise one can not make a decisive conclusion.”

We agree that this is a useful control. We have included polysome profiles lacking emetine or cycloheximide to provide a point of comparison. We find a similar peak for the 80S population with a clear reduction in polysomes (Fig. S3A). We observe monosome accumulation in both polysome profiles (independent of antibiotic) and total ribosomes isolated by SEC. Similar structures were obtained from others following starkly different protocols in the absence of elongation inhibitors (Brown et al. *eLife* 2014, Anger et al *Nature* 2013). We therefore conclude that stabilization of eEF2 and SERBP1 on vacant monosomes is not an artifact of our methodology.

“Why two using different approaches...”

There were several practical concerns that influenced our approach, first, we do not have access to a working fractionator. Our repairs been unsuccessful as our beloved unit is ... a classic model. Second, we had difficulty probing manually isolated fractions for p-eEF2 due the harshness of TCA precipitation and relative insensitivity of our antibody. Our polysome profiles and cryo-EM analysis indicate profound lack of polysomes following nelfinavir treatment. Thus, they would only weakly contribute to any signal from sucrose cushion blots.

“A nice control to have would be a sucrose gradient experiment (just the UV trace) using eEF2K KO cell line treated by Nelfinavir.”

This is an excellent idea that we would really like to incorporate. However, the eEF2K cells are obtained from dissociated adult DRG tissues. As the neurons are terminally differentiated, the neurons are incapable of mitosis. Thus, while we agree that this would be an excellent control, it requires a prohibitive number of animals for biochemical experiments (e.g. total DRGs from ~6 mice per sucrose cushions = 18-22 mice for this control). Thus, this is not practically feasible considering the time and animal numbers associated with generating this size cohort for a control. It is because of this limitation we pre-treated F11 cells with an eEF2K inhibitor before treatment with nelfinavir (Fig. 5A, blue line) as an alternate control.

“...cryo-EM analysis of stable 80S formed upon Nelfinavir treatment. Additional analyses are needed: i) a figure showing local resolution of cryo-EM maps for classes of interest...”

We have added local resolution maps of both eEF2-containing classes and additional analyses to the manuscript as suggested (Fig S5).

ii) zoom-in on the model and density for SERBP1

We have added a zoom-in and the density for SERBP1 to the manuscript as suggested (Fig. 6).

“The focus of the figure is on eEF2-P – but it is possible that the SERBP1 who is doing the work of 80S stabilization (see below), therefore it is essential to document how well resolved it is. Next the authors ask a question of eEF2-P and SERBP1 protect the ribosomes from splitting. For this they use cells purified from Nelfinavir-treated cells. As they show in Figure 5, Nelfinavir treatment results in SERBP1 recruitment to 80S (and stabilizes them?). Therefore, one can not make a decisive conclusion that it is eEF2-P that stabilizes the ribosomes – it could be SERBP1, which would then promote the eEF2-P association (see <https://www.ncbi.nlm.nih.gov/pmc/articles/PMC6226290/>). Therefore, current assays do not necessarily answer the question ‘We therefore asked if phosphorylated eEF2 protects ribosomes from recycling’, it is also possible that instead they ask the question is translational shutoff resulting in SERBP1-mediated 80S stabilization stabilizing the 80S?

At least in yeast i) SERBP1/Stm1p actively inhibits translation and as a consequence of that eEF2-unrelated elongation factor 3, eEF3, is stabilized on the 80S <https://www.ncbi.nlm.nih.gov/pmc/articles/PMC2764444/> and 2) deleting Stm1p-encoding gene makes translation more active (at least in lysates) <https://pubmed.ncbi.nlm.nih.gov/30619132/> . Therefore, my money is on SERBP1 being the driver here - not eEF2-P....

We have carefully edited the manuscript in light of the reviewer’s comments to make it clear that our data do not demonstrate that phosphorylation of eEF2 is the sole factor responsible for the production of the vacant ribosomes.

To the notion about decisiveness surrounding eEF2-P, we have added an additional experiment to clarify if the effects of nelfinavir on splitting are acting through eEF2K/eEF2 (Fig. 7D). We found that pre-treatment with an eEF2K inhibitor diminishes the effect of nelfinavir, providing additional support for our hypothesis.

We would note that a broad range of stimuli that promote formation of vacant ribosomes also stimulate AMPK, an upstream activator of eEF2K. Two examples are glucose deprivation and hypoxia (Long YC JCI 2006, Surks MI, Berkowitz M Am J Physiol 1971). How broad is the role of eEF2K in stabilization of vacant ribosomes? eEF2 inhibits eEF3-catalysed ribosome splitting in yeast although it is unclear if phosphorylation plays a role in these experiments (Kurata S et al NAR 2013). Our study provides a conceptual framework for addressing this problem.

To this end, we agree that to ask if SERBP1 promotes binding of phosphorylated eEF2 to ribosomes we would need to conduct splitting assays with individually purified components. We note this in the discussion. While we agree with the importance of these experiments, we did not undertake them given the tremendous technical challenge of generating sufficient amounts of purified eEF2. Due to the diphthamide modification, it cannot be obtained from bacteria. Thus, this series of experiments would easily constitute the basis of an entirely separate study – one that would be valuable. Our data are entirely consistent with the notion that eEF2 and SERBP1 cooperate to protect 80S ribosomes from recycling, which we elaborate on in the Discussion. The order of this association will be critical to elucidate.

Lastly, we would note that a recent paper (Hayashi H et al. The Journal of Biochemistry 2018) provides evidence in line with our model working in yeast. In this work, Stm1-induced inhibition of translation is antagonized by eEF3, and Stm1 binds along with eEF2. Thus, there is conflict as to which factor is the tail and which one is the dog. For this reason, we are more circumspect on how to place our wager.

Reviewer #2

1. The abstract should make it clear that nelfinivir's effect on p-body abundance is restricted to sensory neurons, as the data shows it has no effect in other cell types (U2-OS).

We have incorporated the reviewer's suggestion.

2. The last paragraph of the introduction is hard to follow and largely redundant with the results section. I suggest revising this paragraph to briefly summarize the take home messages rather than to foreshadow the experiments.

We apologize for the poorly constructed section and have revised it substantially.

3. A figure similar to figure 6E would be useful earlier in the manuscript. It would be especially useful if it included the identity of the key reagents used in the study (nelfinavir, A484954, MK801, etc) and their modes of action.

We have generated several additional schematics to aid the reader in following the logic of our experiments (please see panels A in Figures 2 and 4).

5. It would be helpful to present the mode of MK801 inhibition prior to describing the data. It would help the audience to understand the rationale behind the experiment.

We have improved the set up to this experiment as suggested.

6. What are the implications of the non-rotated conformation and the comparison to rabbit reticulocyte lysate structure described on the bottom of page 10?

Great question - the conformations we observe are virtually identical to the models described by Brown et al. eLife 2014, which we have clarified in the text. The phosphorylation site T56 appears unphosphorylated in the rotated state, the nonrotated state likely represents the phosphorylated state, as judged by the disorder of switch I. As the phosphorylation site is oriented towards the LSU of the ribosome and thus occluding the potential binding site of the kinase, it seems that phosphorylation of eEF2 likely precedes binding to the ribosome. Whether eEF2 spontaneously de-phosphorylates and converts into the rotated state – known to be the preferred orientation of PRE-translocation ribosomes – or whether active dephosphorylation occurs is unclear, though the occlusion of T56 similarly suggests that a phosphatase is likely unable to access the site.

Minor comments:

1. In figure 2A/B, does A484954 lead to a significant increase relative to vehicle? If so, this should be addressed in the text.

The apparent increase was not significant as determined by one-way ANOVA. This has now been noted in the figure.

2. In Fig4 panel B please add NFV only and MK801 only treated blots for comparison.

We do not feel this is necessary as the data are in a proceeding figure.

3. Fig5 panel B. I presume RPL5 and RPS6 are used as large subunit / small subunit loading controls. This should be clarified for the non-experts in the methods section.

We have corrected this as suggested.

4. There is a misspelling in FigS2 legend "FUCAT".

We have corrected this as suggested.

5. What was the rationale for the use of the U2-OS cell line in Fig S2 (as opposed to other non-neuronal lines)?

We made use of this line as it is widely used to study RNP granules (e.g. refs. 13, 21, 68, Marmor-Kollet et al. Mol Cell 2020, Gwon et al. Science 2020, Jayabalan et al. Nat Comm 2016, etc.). This is provided as a point of comparison for the notion that the molecular mechanisms responsible for p-body number differ between mitotically active cells and terminally differentiated sensory neurons (Fig. 1B). We have clarified our rationale in the text.

Reviewer #3

"...the role of PBs generally is not well understood in any system. So by choosing studying them, authors should have a physiological relevance especially in their model, sensory neurons. Although they provide a general model as a part of figure 6E, I do see how PBs in general and then recycling part fits to the obtained results."

We agree that the biological purpose of p-bodies is poorly understood. We feel that this is a compelling reason for why it is worthwhile to examine them. A potential way to illuminate their function is understanding their relationship to well understood components of the translation cycle. We identified a new molecular mechanism that links translation to neuronal p-bodies and we feel that this is a valuable contribution towards deciphering how they are controlled. A striking difference between sensory neurons and cell lines is that there are substantially more p-bodies in the former. While it is purely speculative, this may reflect added importance in certain cell types – we simply don't yet know what their biological function might be and this is stated in the manuscript. However, it is clear that early models which imply a zero-sum game between p-bodies and translation do not capture the situation in sensory neurons.

The connection between SNPB abundance and translation appears to be eEF2K activity. In probing how eEF2K activity controls translation, we found that the molecular mechanism that underlies translational arrest in our system may involve stabilization of vacant ribosomes. This is important because they are widely observed but the mechanisms that govern their genesis are poorly understood. We feel that this work provides valuable insights into molecular mechanisms that influence mRNA control.

"The work is fragmented and has no solid molecular mechanism data connecting two possible

phenomena : formation of PBs and regulation of ribosome availability via eEF2K-ph-eIF2-SERBP1 axis.”

We can appreciate that this work integrates a series of complex pathways and we attempted to make the story as cohesive as possible. We disagree that we do not provide data that connect eEF2K activity to p-bodies and translation. We would highlight as examples figures 2B, 2C, 4B, 4C, 4D, 5A, 5B, and 7B-E. The data span a remarkable range of resolution from whole cell imaging to atomic models – we view the multi-pronged approach as yielding a solid body of evidence in support of our central thesis regarding the critical (and novel) functions of eEF2K/eEF2 signaling.

“The cryo-em part is solid but has been published earlier in other systems by other laboratories. While structural data is convincing, it does not provide any type of real quantification in vivo or in cells/neurons. What is the proportion of the ribosomes that become associated with SERBP1 upon nelfinavir treatment? I doubt that the assay based on the relative splitting can result in the adequate and unbiased quantification of such events.”

We agree that similar structures have been reported and we cite this work in the manuscript. We note that a critical difference between our study and what is known is that we are able to contextualize the structure – we describe factors that promote accumulation of the vacant ribosomes (changes in eEF2K activity), we describe the structural consequences of eEF2 phosphorylation, and importantly we describe the consequences of eEF2K on recycling. The level of resolution we were able to obtain enabled us to pursue this hypothesis which is also distinct from the reported structures. Finally, the fact that these are observed in other systems suggests that analogous mechanisms could come into play elsewhere and is a major strength of our work.

To the second point, approximately 70% of the ribosomes isolated from primary DRG neurons following nelfinavir treatment are devoid of mRNA and bound to SERBP1 and eEF2. In other unpublished datasets we typically find that ~20-25% of ribosomes are bound to SERBP1 and eEF2, though we avoided direct comparisons as these ratios, though unlikely for ribosomes, can be altered due to uncontrollable variability in specimen preparation (eg. complexes can fall apart if exposed too long to the air-water interface - during specimen preparation the sample is suspended in a thin buffer film, which increases the surface area and evaporation is difficult to control. Some complexes might be more prone to dissociate). To allow for a comparison between baseline and nelfinavir treated samples, we included polysome gradients in the manuscript that indicate the relative amount of 80S ribosomes (Fig. 5A).

Finally, we were careful in the manuscript not to suggest that splitting assays measure SERBP1 association.

“The manipulation with KO cells are interesting but need rescue experiments with WT or even better rescue experiments with eIF2K mutants.”

While we feel rescue experiments would be of interest in the KO cells, they are not tractable for technical reasons. The eEF2K neurons are obtained from dissociated adult DRG tissues. This system is refractory to standard approaches for transfection. Should this become a feasible approach, it is absolutely one that we would be interested in applying.

“It is not clear why authors used only one single PB marker, RCK. In fact this marker is also SG marker. Moreover, the quality of IF pictures both for PB quantification and AHA incorporation is very low and needs better resolution.”

We share the reviewer's concern that the selection of antibodies for our imaging studies is critically important. We selected to use RCK for the following reasons:

- RCK is enriched in PBs (see ref. 12). We were unable to find manuscripts where it was used as a marker for SGs.
- RCK is a well-established marker for p-bodies (see refs. 20, 60-62, Wilbertz JH et al *Mol Cell* 2019, Ayache J et al *MBC* 2015, Kamenska A *NAR* 2016, Kedersha N and Anderson P *Methods Enzymol.* 2017, etc.).
- Neuronal PBs and SGs differ in that PBs are not present in axons (see refs 17-19), while SG markers do form aggregates in axons (Sahoo et al *Nat Comm* 2018). Consistent with this, we observe that RCK puncta are excluded from axons in primary DRG neurons.
- The morphology of SGs can be quite distinct from PBs, the former often being fewer in number and much larger than the latter. And, unlike PBs, SGs are not constitutively present. We consistently observe abundant RCK puncta in vehicle-treated neurons (see Figs. 1, 2, 4), further suggesting we are measuring PBs.
- One of our key findings is that nelfinavir results in a drastic reduction in RCK puncta. Stressors that induce SGs are also known to increase PB numbers. It is therefore unlikely that the eEF2K-dependent reduction in PBs is affected by SGs.

As an additional experimental validation, we examined co-localization of RCK with a second marker - Dcp2. In the representative image (right), green and red denote DCP2 and RCK, respectively, and white indicates colocalization of the two signals. The vast majority of puncta show colocalization of RCK and DCP2 (with some exceptions, examples indicated by arrows), consistent with the presence of p-bodies.

To the second point regarding resolution, as indicated by the quantification presented in the imaging experiments, our controls establish that our resolution is more than able to detect significant changes in translation and SNPBs. While we would like to provide larger images for each of the representative images that would improve the clarity of the data in the manuscript, we are limited by space constraints imposed by the volume of data contained in the manuscripts and constraints on the number of supplemental items allowed by the journal.

"The use of chemical inhibitors here such as MK801 that has several other pleiotropic effects besides NMDAR-related effects is unreliable."

We can appreciate the reviewers concern that the specificity of pharmacologic manipulations is critical. Our approach for probing NMDAR function was to seek out the best possible tool. We found that MK-801 has been cited 7,899 times with the majority of the citations capitalizing on its well-established properties as an NMDAR antagonist. Thus, we agree that while specificity is a concern for some drugs, the ubiquitous use of MK-801 in this context suggests that it is widely regarded as useful.

"It is also not clear why authors continued to study coupling between translation and SNBPs, if they conclude (Fig 1, and page 6) that "the ongoing protein synthesis and SNBP abundance are not directly coupled. Based on these result, we reasoned that factors involved in translation elongation might play critical roles in coordinate regulation of translation and SNBPs". This is puzzling!"

We apologize for the confusion caused by this statement. What we were attempting to convey is that the relationship between translation and p-bodies is nuanced. This is in contrast to other biological contexts where it has been investigated (we have highlighted these studies in the introduction section). We are intrigued that a specific elongation factor kinase appears to integrate the two processes. This was not our initial expectation based on the available literature.

"Another aspect of the study that is critical is that authors do not show any dose- and time-response for any given experiment...."

We selected our drug concentrations based on previously reported values in the literature. While we agree that additional dose and time studies would be ideal, we feel that our data provide ample evidence in support of the claims we make in the manuscript. However, in acknowledgement of the reviewers' concern, we now note in the discussion that a potential caveat to our measurements is that we did not test a wide range of concentrations and timepoints.

Please do not hesitate to contact us for any additional information. And thank you.

REVIEWERS' COMMENTS

Reviewer #1 (Remarks to the Author):

I am satisfied with the revision. Given the technical limitations (lack of functioning gradient station, prohibitive numbers of mice necessary for performing some of the requested experiments etc.) I believe this is the best the authors could do. I recommend acceptance at this stage.

Reviewer #2 (Remarks to the Author):

The revised manuscript is acceptable for publication. All of my previous concerns have been fully addressed in this revision.

Reviewer #3 (Remarks to the Author):

The manuscript is significantly improved. I think that most of my concerns were adequately addressed. I can now recommend this work for the publication.

The reviewers recommended acceptance and did not request additional changes. These are the comments from the last round of review:

Reviewer #1 (Remarks to the Author):

I am satisfied with the revision. Given the technical limitations (lack of functioning gradient station, prohibitive numbers of mice necessary for performing some of the requested experiments etc.) I believe this is the best the authors could do. I recommend acceptance at this stage.

Reviewer #2 (Remarks to the Author):

The revised manuscript is acceptable for publication. All of my previous concerns have been fully addressed in this revision.

Reviewer #3 (Remarks to the Author):

The manuscript is significantly improved. I think that most of my concerns were adequately addressed. I can now recommend this work for the publication.

We are grateful for their comments which resulted in a stronger manuscript.